# Using $\delta^{13}$C-CH$_4$ and $\delta$D-CH$_4$ to constrain Arctic methane emissions

Nicola J. Warwick[1,2], Michelle L. Cain[1], Rebecca Fisher[3], James L. France[4], David Lowry[3], Sylvia E. Michel[5], Euan G. Nisbet[3], Bruce H. Vaughn[5], James W. C. White[5] and John A. Pyle[1,2]

[1]National Centre for Atmospheric Science, NCAS, UK.
[2]Department of Chemistry, University of Cambridge, Lensfield Road, Cambridge, CB2 1EW, UK.
[3]Department of Earth Sciences, Royal Holloway, University of London, Egham, TW20 0EX, UK.
[4]School of Environmental Sciences, University of East Anglia, Norwich, NR4 7TJ, UK.
[5]Institute of Arctic and Alpine Research (INSTAAR), University of Colorado, Boulder, CO 80309, USA.

*Correspondence to*: Nicola J. Warwick (Nicola.Warwick@atm.ch.cam.ac.uk)

**Abstract.** We present a global methane modelling study assessing the sensitivity of Arctic atmospheric CH$_4$ mole fractions, $\delta^{13}$C-CH$_4$ and $\delta$D-CH$_4$ to uncertainties in Arctic methane sources. Model simulations include methane tracers tagged by source and isotopic composition and are compared with atmospheric data at four high northern latitude measurement sites. We find the model's ability to capture the magnitude and phase of observed seasonal cycles of CH$_4$ mixing ratios, $\delta^{13}$C-CH$_4$ and $\delta$D-CH$_4$ in high northern latitudes is much improved using a later spring kick-off and autumn decline in high northern latitude wetland emissions than predicted by most process models. Results from our model simulations indicate that recent predictions of large methane emissions from thawing submarine permafrost in the East Siberian Arctic Shelf region could only be reconciled with global scale atmospheric observations by making large adjustments to high latitude anthropogenic or wetland emission inventories.

## 1 Introduction

Methane is an important greenhouse gas that has more than doubled in atmospheric concentration since pre-industrial times. Following a slow-down in the rate of growth in the late 1990s, the methane content of the atmosphere began increasing again in 2007 (Dlugokencky et al., 1998, Bousquet et al., 2011, Nisbet et al., 2014). Although this increase has occurred globally, latitudinal differences in methane growth rates suggest multiple causes for the renewed growth. In 2007, the Arctic experienced a rapid methane increase, but in 2008 and 2009-10 growth was strongest in the tropics. This renewed global increase in atmospheric methane has been accompanied by a shift towards more $^{13}$C-depleted values, suggesting that one explanation for the change could be an increase in $^{13}$C-depleted wetland emissions (Nisbet et al., 2016). However, other factors such as changing emissions from ruminant animals (Schaefer et al., 2016) and the fossil fuel industry could also play a role (Bergamaschi et al., 2013, Kirschke et al., 2013, Hausmann et al., 2016).

The Arctic contains important methane sources that are currently poorly quantified and climate sensitive, with the potential for positive climate feedbacks. The largest and most uncertain of these are emissions from wetlands (e.g. Melton et al., 2013, Saunois et al., 2016). While wetland methane fluxes can be obtained experimentally by chamber studies and eddy correlation

techniques (e.g. Pelletier et al., 2007, O'Shea et al, 2014), the heterogeneous conditions in wetlands and seasonal and interannual variation in wetland area (Petrescu et al., 2010) can lead to large uncertainties, both spatially and temporally, when upscaling this data. As high latitude wetland emissions are generally considered to occur from May melt to October freeze-up (Bohn et al., 2015, Christensen et al., 2003), and due to difficulties conducting field campaigns during the winter and spring-melt seasons, to-date most experimental Arctic wetland flux data has been reported for the summer season. However, a recent Arctic wetland study using year-round eddy flux data reported the presence of large methane emissions continuing well into winter, when subsurface soil temperatures remain close to 0°C (Zona et al., 2016). This study concluded that cold season (September-May) fluxes dominated the Arctic tundra methane budget.

Methane emissions from wetlands can also be estimated using process-based models. However, a recent model intercomparison study, WETCHIMP, showed wide disagreement in the magnitude of global and regional emissions among large-scale models (Melton et al., 2013). The magnitude of methane emissions from high northern latitude wetlands (>50° N) varied from 21 to 54 Tg yr$^{-1}$ (Melton et al., 2013), representing approximately 5 to 10 % of the total global methane emission budget. There was also significant variability between models in the seasonal distribution of these emissions. Figure 1 shows a comparison of seasonal cycles of high northern latitude wetland emissions from the WETCHIMP models, the wetland dataset described in Fung et al. (1991) and the model inversion study of Bousquet et al. (2011). There is significant spread in how the emissions are distributed throughout the year, with the summertime peak in emissions occurring in June, July or August depending on the model considered. In a model intercomparison focusing on wetland emissions in West Siberia (WETCHIMP-WSL, Bohn et al., 2015), the largest disagreement in the temporal distribution of emissions occurs in springtime (May and June). During this period, the range in normalised model monthly emissions spans from a minimum of negative values (representing methane uptake) to a peak in the emission seasonal cycle. This large uncertainty associated with the timing of, and processes controlling, seasonal variations in wetland methane emissions needs to be resolved before predictions can be made of how emissions might change in a changing climate.

Decomposing gas hydrates may also represent a small, but significant, climate sensitive methane source. Shallow methane hydrates in Arctic regions may be particularly vulnerable to destabilisation following increases in temperature as a result of climate change. Furthermore, thawing permafrost could release methane previously trapped below in shallow reservoirs, including hydrates, to the atmosphere. Previous studies of the methane budget have either omitted a hydrate source or used a global value for Arctic hydrate emissions of 5 Tg yr$^{-1}$. However this value is no more than a placeholder suggested by Cicerone and Oremland (1988). More recently, Shakhova et al., (2010) and Shakhova et al. (2014) used ship-based observations to estimate methane emissions from thawing permafrost on the East Siberian Arctic Shelf (ESAS). They estimated a total ESAS methane source from diffusion, ebullition and storm-induced release from subsea permafrost and hydrates of 17 Tg yr$^{-1}$; significantly more than the 5 Tg yr$^{-1}$ suggested by Cicerone and Oremland (1988). However, a recent study by Berchet at al. (2016) using an atmospheric chemistry transport model, found that an ESAS source as high as 17 Tg yr$^{-1}$ was inconsistent with atmospheric observations of methane mole fractions at northern high latitude measurement sites. In the Berchet et al. (2016)  study, ESAS emissions were estimated to be in the range 0.5 to 4.3 Tg yr$^{-1}$.

Other recent studies identifying additional potential northern high latitude sources and sinks of methane include emissions from Arctic thermokarst lakes (11.86 Tg yr$^{-1}$, Tan and Zhuang, 2015), polymers in oceanic ice (~7 Tg yr$^{-1}$, Kort et al., 2012) and methane uptake by boreal vegetation (~-9 Tg yr$^{-1}$, Sundqvist et al., 2012). These studies have either used process-based models or extrapolated local observations to calculate Arctic fluxes that would all be highly significant on a regional scale.

However, uncertainties in these sources are high as many fluxes may be episodic as well as spatially scattered, and could therefore be missed by relatively infrequent field campaigns. In addition to natural sources, the Arctic contains methane emissions from some of the world's largest gas producing plants, situated in northern Russia (Reshetnikov et al., 2000; EDGAR v4.2, http://edgar.jrc.ec.europa.eu, 2011).

The main atmospheric sink of methane is reaction with the hydroxyl radical, OH. Other lesser sinks include reaction with Cl

in the boundary layer (e.g. Allan et al., 2007, Lawler et al., 2011, Banton et al., 2015), reaction with Cl and O($^1$D) in the stratosphere and uptake of methane by methanotrophs in oxic soils. These sinks all vary seasonally due to seasonal changes in solar insolation and temperature etc.. Overall, knowledge of source and sink partitioning within the Arctic methane budget is poor, and a better understanding of emissions is required to determine the best emission reduction strategies and feedbacks in a future climate.

Along with atmospheric modelling, measurements of methane mole fractions provide important information on the geographic and seasonal distribution of methane emissions. However, mole fraction measurements alone do not give us the ability to distinguish between emissions from different methane sources. This can be achieved in a broad sense using observations of stable isotope ratios in methane as different sources have distinct isotopic ratios. For example, methane emitted from wetlands is relatively more depleted in $^{13}$C than that from fossil sources, which are in turn depleted relative to

methane derived from biomass burning (Dlugokencky et al. 2011). To date, global atmospheric modelling studies have only incorporated information on the $^{13}$C/$^{12}$C ($\delta^{13}C_{CH4}$) composition of methane using geographically uniform source isotopic signatures. However, new information on the atmospheric distribution of the D/H composition (White et al., 2016) provides an additional potential discriminant between source and sink strengths. Rigby et al. (2012) included both $^{13}CH_4$ and $CH_3D$ tracers in an atmospheric model to quantify uncertainty reductions in future methane emission estimates that could be

achieved if measurement networks performed high-frequency and precision isotopic measurements. However, model results were not compared to existing atmospheric isotopic data in this study. Here we present the first modelling study of modern methane to (a) include published large geographical variations in the isotopic signature of wetland emissions and (b) assess methane emission scenarios against atmospheric observations of $\delta D_{CH4}$.

Global model simulations are performed using the p-TOMCAT 3D chemistry transport model using offline chemistry

(Warwick et al., 2006) and multiple methane tracers tagged by source and $\delta^{13}$C and $\delta$D isotopic composition. We investigate the sensitivity of atmospheric distributions of $CH_4$, $\delta^{13}C_{CH4}$ and $\delta D_{CH4}$ to changes in fluxes from climate-sensitive Arctic sources and analyse potential causes of differences between models and measurements in this region.

## 2 Measurements

Model results are compared to monthly mean weekly flask observations of $CH_4$ mixing ratios, $\delta^{13}C\text{-}CH_4$ and $\delta D\text{-}CH_4$ from NOAA-ESRL sampling sites at Alert (82°N, 63°W), Ny-Alesund (79°N, 12°E), Barrow (71°N, 157°W) and Cold Bay (55°N, 163°W) (Dlugokencky et al., 2013; White and Vaughn, 2015; White et al., 2016). These sites were selected for comparison as they are the four most northerly sites with simultaneous $CH_4$, $\delta^{13}C\text{-}CH_4$ and $\delta D\text{-}CH_4$ observation data. In addition, modelled latitudinal gradients of $CH_4$, $\delta^{13}C\text{-}CH_4$ and $\delta D\text{-}CH_4$ are analysed by comparison with annual mean observations from a further 8 NOAA-ESRL sampling sites spread over latitudes 90° S to 53° N. The location of these measurement sites is shown in Figure 3 (due to the proximity of the measurement sites at Mauna Loa and Cape Kumukahi they appear as one point). Monthly mean observations are averaged over the years 2005 to 2009 (the period for which there is $\delta D\text{-}CH_4$ data available). NOAA-ESRL was responsible for the collection of the sample and logistics, with cooperating agencies. Samples were then analysed for methane mixing ratios at NOAA-ESRL in Boulder, Colorado, with an analytical repeatability of 0.8 to 2.3 ppb. Stable isotopic compositions were determined at the Stable Isotope Laboratory at INSTAAR, part of the University of Colorado, Boulder, with a precision of better than 0.1 ‰ for $\delta^{13}C\text{-}CH_4$ (White et al., 2015; Miller et al., 2002) and 2‰ for $\delta D\text{-}CH_4$ (White et al., 2016).

## 3 Isotopic composition of methane

The isotopic composition of atmospheric methane is generally expressed in 'delta' notation, as the isotopic ratio in the sample compared to an international standard. The original standard for the $^{13}C/^{12}C$ ratio was Pee Dee Belemnite, a fossil from the Pee Dee marine carbonate formation in South Carolina (Craig, 1957), which established the V-PDB scale. For the D/H ratio, the international standard is Vienna Standard Mean Ocean Water (VSMOW) (DeWitt et al., 1980). The delta values for the two main stable isotopologues of methane are given by

$$\delta^{13}C = 1000\left(\frac{R_{13CH4}}{R_{PDB}} - 1\right) \tag{1}$$

$$\delta D = 1000\left(\frac{R_{CH3D}}{R_{VSMOW}} - 1\right) \tag{2}$$

where $R_x$ is the molar ratio of $^{13}C$ or D to the most abundant isotopologue (i.e. $^{12}C$ or H respectively). $R_{PDB}$ is the $^{13}C/^{12}C$ ratio found in V-PDB and $R_{VSMOW}$ is the D/H ratio found in V-SMOW. Global mean surface atmospheric observations of $CH_4$, $\delta^{13}C\text{-}CH_4$ and $\delta D\text{-}CH_4$ were ~1780 ppb,~-47.2 ‰ and ~-86 ‰ respectively for the 2005 to 2009 period (Dlugokencky et al., 2013; White and Vaughn, 2015; White et al., 2016). Geographical and altitudinal variations in these compositions arise as a result of variations in the distributions of the isotopic composition of the parent organic matter, the method of production (pyrogenic, thermogenic or biogenic) and differing rates of destruction between methane isotopologues. At large scales, the $\delta D$ composition of methane is controlled by the $\delta D$ of water present, while at smaller scales, the methods of production and destruction may play a more important role. Likewise the $\delta^{13}C$ composition of methane can be influenced by the type of parent organic matter (e.g. C3 or C4 vegetation), as well as the method of production. As different methane

sources tend to have distinct isotopic ratios, observations of the isotopic composition of atmospheric methane can be used as additional constraints on the methane budget (e.g. Rigby et al., 2013; Schaefer et al, 2016).

## 4 Model description

The global 3D chemical transport model, p-TOMCAT, has been used extensively for tropospheric studies and is described in more detail in Cook et al. (2007) and Warwick et al. (2013). For this study, the model was run at a horizontal resolution of ~2.8° x 2.8°, with 31 levels extending from the surface to 10 hPa. The horizontal and vertical transport of tracers was based on 6-hourly meteorological fields, including winds and temperatures derived from the operational analyses of the European Centre for Medium Range Weather Forecasts (ECMWF) for 2009.

The version of p-TOMCAT used in this work has been modified to include parameterised chemistry where tagged-source-type methane tracers of $^{12}CH_4$ and $^{13}CH_4$, and a 'total' $CH_3D$ are destroyed via reaction with OH, $O(^1D)$ and Cl. The OH distributions are prescribed hourly values taken from a full chemistry version of p-TOMCAT and compare well with other global OH distributions described in the literature, giving a global methane lifetime of 10.4 years with respect to OH (for more details see Warwick et al., 2006). A comparison of modelled seasonal cycles of methyl chloroform and observational data from the NOAA-ESRL halocarbons in situ program at Barrow, Alaska, suggests that the seasonal cycle of the model prescribed OH concentrations is well represented in the Arctic region (see Fig. S1). Although there is a slight difference in the timing of the observed and modelled methyl chloroform minima, the modelled seasonal cycle falls well within the range of observations. The stratospheric destruction of methane by reaction with Cl and $O(^1D)$ is derived from prescribed 2D Cl and $O(^1D)$ 5 day mean distributions taken from the Cambridge 2D model (Bekki and Pyle, 1994). Mixing ratios of Cl in the marine boundary layer are prescribed with latitudinal and seasonal variations according to Allan et al., (2007). Global atmospheric methane lifetimes with respect to the Cl and O(1D) stratospheric and Cl marine boundary layer reactions are 265 and 360 years respectively. Reaction rate coefficients for the reaction of $CH_4$ with OH is taken from Burkholder et al. (2015) , and with O(1D) and Cl from Atkinson et al. (2004). Kinetic isotope effects (KIEs, defined as the ratio of rate constants for the reactions involving the reactant and an isotopically substituted reactant with a certain species) for the methane reaction rates are included in the model chemistry scheme and are listed in Table S1. Oxidation of methane by soils is treated as a negative emission following Fung et al., (1991).

Methane emissions and source-specific isotopic signatures used in the p-TOMCAT BASE control scenario are described in Table 1. The geographical and seasonal distribution of methane fluxes are taken from EDGAR v4.1 (http://edgar.jrc.ec.europa.eu/overview.php?v=41) for 2005, Fung et al. (1991) and Van der Werf et al. (2006). The geographical distribution of wetland emissions above 50° N is shown in Fig. 2. Further details on the fluxes and source-specific isotopic signatures used in the model are outlined in the Supplementary Online Material. Methane tracers of $^{12}CH_4$, $^{13}CH_4$ and $CH_3D$ are tagged by source type as shown in Table 1. In addition, the 'Northern Wetlands' tracer is also tagged by continental region, with emission regions split into North American, North European or north Asian. Different emission and

sink scenarios considered in this study and their variations from the BASE scenario are described in later sections and listed in Table 2.

Initially, a 'total' methane tracer was spun-up in a 40-year single-tracer simulation until calculated year-to-year changes in local methane mole fractions were negligible. The tagged methane source tracers in each scenario were then initialised by scaling this spun-up total methane tracer globally, according to the global emission fraction and isotopic composition of the source. Results presented here are taken from the final year of further 40-year simulations using perpetual 2009 meteorology, after which year-to-year changes in the local mole fractions of the individual tracers were deemed to be negligible (<0.5 %), along with the associated changes in $\delta^{13}$C-CH$_4$ and $\delta$D-CH$_4$.

## 5 Atmospheric distribution of methane mole fraction and isotopic composition

### 5.1 Global distribution

Figure 3 shows the modelled annual mean surface distributions of total CH$_4$, $\delta^{13}$C-CH$_4$ and $\delta$D-CH$_4$ for the BASE scenario. The results are broadly comparable to observational data, with higher mixing ratios and lighter (more negative) isotopic fractionations occurring in the Northern Hemisphere (NH) than the Southern Hemisphere (SH). This gradient in isotopic fractionations arises as the rates of reaction of OH, Cl and O($^1$D) with $^{13}$CH$_4$ and CH$_3$D are all fractionally slower than with $^{12}$CH$_4$ (see Table S1). Therefore, both $\delta^{13}$C and $\delta$D increase (become more enriched in the heavy isotope) with increased exposure to atmospheric sinks. As the majority of methane emissions are located in the NH, and because these are predominantly depleted in heavy isotopes, there are strong latitudinal gradients in methane and its isotopic fractionations: higher concentrations and more negative $\delta^{13}$C-CH$_4$ and $\delta$D-CH$_4$ values are found in the NH than the SH. Regional variations in $\delta^{13}$C-CH$_4$ and $\delta$D-CH$_4$ also occur due to regional variations in methane source types with differing isotopic signatures (see Table 1).

The model captures the observed latitudinal gradients in CH$_4$, and $\delta^{13}$C-CH$_4$ (see Fig. 4). The latitudinal gradient in $\delta$D-CH$_4$ is also well represented, except for a step change between the South Pole and lower southern latitudes in the observations that is not captured by the model. One reason for this could be errors in the model scenario. However, given the well-mixed nature of both SH CH$_4$ mixing ratios and $\delta^{13}$C-CH$_4$ values, the limited amount of $\delta$D-CH$_4$ data available, and the precision of the measurements, it is also possible that this step change in the SH latitudinal gradient maybe due to noise in the measurement data.

The latitudinal gradients of CH$_4$, $\delta^{13}$C-CH$_4$ and $\delta$D-CH$_4$ are likely to be strongly influenced by the representation of Arctic methane sources, particularly high latitude wetland emissions, which will give a strong isotopic atmospheric signal due to their very negative $\delta^{13}$C-CH$_4$ and $\delta$D-CH$_4$ values. The sensitivity of the modelled latitudinal gradient to variations in particular Arctic methane sources is discussed in more detail in Sect. 6.

## 5.2 Arctic seasonal cycles

### 5.2.1 Comparison of the base simulation with observations

The observed seasonal cycle of $CH_4$ mole fractions in high northern latitudes is dominated by a sharp summer minimum in July, and a broader winter maximum from October to March (see Fig. 5). This seasonal cycle arises as a result of seasonal

variations in the major methane sink, reaction with OH, seasonal variations in the surface sources of methane and seasonal changes in vertical mixing and horizontal transport. For example, the Arctic is influenced by long-range transport of airmasses containing high levels of anthropogenic methane from lower latitudes during winter and spring (e.g. Dlugokencky et al. 1995; Worthy et al. 2009). Model studies have had difficulty capturing seasonal cycles of methane in high northern latitudes (e.g. Houweling et al., 2000; Wang et al., 2004; Pickett-Heaps et al., 2011), in particular the timing of the summer

minimum.

Observed seasonal cycles of $\delta^{13}C$-$CH_4$ and $\delta D$-$CH_4$ show some level of anti-correlation with $CH_4$ mole fractions. If the observed seasonal cycle of $CH_4$ were due to reaction with OH alone, then the KIEs of the $CH_4$ + OH reaction would result in $\delta^{13}C$-$CH_4$ and $\delta D$-$CH_4$ seasonal cycles 180° out of phase with the $CH_4$ seasonal cycle: the minimum in $CH_4$ mixing ratio corresponding to maxima in $\delta^{13}C$-$CH_4$ and $\delta D$-$CH_4$. However, phase relationships between observed seasonal cycles in $CH_4$,

$\delta^{13}C$-$CH_4$ and $\delta D$-$CH_4$ are also influenced by seasonal variations in surface sources and lesser, alternate sinks leading to more complicated phase relationships.

The reaction of $CH_3D$ with OH has a larger KIE than the reaction of $^{13}CH_4$ with OH (see Table S1). Therefore seasonal variations in atmospheric $\delta D$-$CH_4$ will tend to be more dominated by seasonal changes in the OH sink than $\delta^{13}C$-$CH_4$, with atmospheric $\delta^{13}C$-$CH_4$ being relatively more influenced by sources. Figure 5 shows that the observed seasonal cycle of $\delta D$-

$CH_4$ is approximately anti-correlated with $CH_4$, as would be expected for a seasonal cycle controlled by seasonal variations in OH. However, this is not true for $\delta^{13}C$-$CH_4$. There is an offset between the $CH_4$ and $\delta^{13}C$-$CH_4$ seasonal cycles, with a period in late spring where $CH_4$ decreases and there is either no change or a slight decrease in $\delta^{13}C$-$CH_4$. In addition, a simultaneous increase in both observed $CH_4$ and $\delta^{13}C$-$CH_4$ from October through to the end of the year demonstrates that factors other than seasonal variations in OH play a role in determining the seasonal cycle of $\delta^{13}C$-$CH_4$.

Figure 5 also shows a comparison of modelled seasonal cycles of $CH_4$, $\delta^{13}C$-$CH_4$ and $\delta D$-$CH_4$ from the BASE scenario with observational data from four high northern latitude sites. Although the model captures the phase and magnitude of observed seasonal cycles in lower northern latitudes (e.g. Cold Bay), clear differences in the magnitude and/or phase are evident in higher latitudes (Alert, Ny-Alesund, Barrow). Analysis of the regionally tagged tracers for wetland emissions >50°N (North American, North European and North Asian), indicate that modelled seasonal cycles at all four measurement sites are

predominantly influenced by American, and to a lesser extent European wetland emissions, with little sensitivity to Asian wetland emissions. The model is unable to capture the magnitude and timing of the Arctic summer minimum in $CH_4$ mixing ratios, while the modelled summer decrease in $\delta^{13}C$-$CH_4$ and $\delta D$-$CH_4$ occurs earlier than observed. In addition, the model underestimates the amplitude of the observed Arctic seasonal cycle in $\delta D$-$CH_4$. These discrepancies point to errors in the

representation of Arctic methane sources and/or the isotopic signature data used within the model, particularly in American and/or European regions. In Sect. 6, we investigate the sensitivity of modelled seasonal cycles to uncertainties in the δD KIE for the $CH_4$ + OH reaction, as well as adjustments in the phase and magnitude of certain Arctic sources.

## 6 Model sensitivities to Arctic source magnitudes and δD isotopic signatures and fractionations

### 6.1 Model sensitivity to KIE$^{H/D}$ and the wetland δD signature

Although the model is able to capture the phase and magnitude of observed seasonal cycles of methyl chloroform in the Arctic, suggesting that the OH seasonal cycle is well represented (Fig. S1), the model underestimates the amplitude of Arctic seasonal cycles of both $CH_4$ and δD-$CH_4$ (Fig. 5). In two further separate model simulations, we investigated the sensitivity of Arctic modelled seasonal cycles in δD-$CH_4$ to (a) uncertainities in the KIE of the $CH_3D$ + OH reaction and (b) uncertainties in the δD signature of methane emissions from high northern latitude wetlands.

Literature KIE values for $k^{CH4+OH}/k^{CH3D+OH}$ range from 1.16 to 1.3, clustering at the higher end of range (DeMore et al., 1993; Gierczak et al., 1997; Bergamaschi et al., 2000; Saueressig et al., 2001; Tyler et al., 2007). In a separate model simulation run parallel to the BASE simulation (DEC_KIE), we find that altering the KIE$^{CH3D+OH}$ reaction within the literature range has an important impact on modelled global mean δD-$CH_4$ values. However, we found the impact of varying KIE$^{CH3D+OH}$ on the magnitude of the modelled δD-$CH_4$ seasonal cycle to be negligible, offering no improvement over the BASE scenario when comparing with observations.

While there is now an increasing amount of data on $^{13}C/^{12}C$ source ratios, D/H ratios for methane sources have been less comprehensively studied and are therefore subject to larger uncertainties. Literature estimates of the δD-$CH_4$ isotopic signature from high northern latitude wetlands range from approximately -300 ‰ to -450 ‰ (e.g. Kulmann et al., 1998; Quay et al, 1999; Nakagawa et al., 2002; Umezawa et al., 2012). However, bulk regional δD values for western Siberian emissions estimated by Yamada et al. (2005) (-482 ‰ to -420 ‰, including the major wetland and fossil fuel sources) suggest a more negative δD signature for wetlands than determined by other studies. Here, in an additional simulation (WETLD_δD), we found that increasing the isotopic signature of >50° N wetland emissions from -360 ‰ to -500 ‰ improved the ability of the model to capture the magnitude of the observed seasonal cycle and latitudinal gradient of δD-$CH_4$ (not shown). However, using such a negative δD signature for high northern latitude wetland emissions would obviously shift the model global mean δD-$CH_4$ to more negative values, and would therefore have to be balanced by further altering the source/sink scenario. In addition, while altering the δD wetland source signature improves the representation of the modelled δD-$CH_4$ seasonal cycle, it does not impact the differences between the modelled and observed $CH_4$ seasonal cycles.

## 6.2 Model sensitivity to the wetland source

### 6.2.1 Varying the source magnitude

Emissions from high northern latitude wetlands (>50° N) are assigned a highly [13]C-depleted and D-depleted isotopic signature (~-70 ‰ and ~-360 ‰ respectively) in the model, as well as a strong seasonal cycle, peaking during the NH summer. Therefore reducing methane emissions from high latitude wetlands in early summer could potentially improve the comparison between observed and modelled seasonal cycles of $CH_4$, $\delta^{13}C$-$CH_4$ and $\delta D$-$CH_4$. Figure 6 shows the influence of varying the magnitude of the wetland source above 50° N on the phase and magnitude of modelled high latitude NH $CH_4$ and $\delta^{13}C$-$CH_4$ seasonal cycles. No results for $\delta D$-$CH_4$ are shown as $CH_3D$ was not tagged by source in the model due to computer integration time limitations.

When the tagged high northern latitude (>50° N) wetland methane tracer (with emissions of 30 Tg yr$^{-1}$) is excluded from the model simulation (NO_WETLD scenario), the summer minimum in $CH_4$ mole fraction occurs later in the year (August/September) than in the BASE scenario and the seasonal variation in $\delta^{13}C$-$CH_4$ is substantially reduced (Fig. 6). When high northern latitude wetland emissions are increased by 50 % (i.e. the annual source strength is increased to 45 Tg yr$^{-1}$, INC_WETLD scenario), the summer minimum occurs earlier in the year (May/June) and seasonal variations in both $CH_4$ and $\delta^{13}C$-$CH_4$ increase relative to the BASE scenario (Fig. 6). Neither wetland scenario provides any improvement in the model's ability to capture observed seasonal cycles: the comparison with observations is worse when high northern latitude wetland emissions are removed, and there are only small changes to model results when high northern latitude wetland emissions are increased by 50 %. We found that altering the Fung et al. (1991) emission distribution in a simple way by varying the relative strengths of the three regional high northern latitude wetland tracers (North American, North European and North Asian) offered no improvement in the agreement between modelled and observed atmospheric seasonal cycles. Modelled seasonal cycles at the measurement station locations showed little sensitivity to emissions from north Asia (including Siberia, see Section 5.2.1), and increasing/decreasing the emission contribution from North America and Northern Europe gave similar results to the INC_WET and NO_WET scenarios.

Figure 7 shows the influence of varying the strength of wetland emissions above 50° N on the modelled latitudinal gradients of $CH_4$ and $\delta^{13}C$-$CH_4$. Removing this source completely dramatically reduces the ability of the model to capture observed latitudinal gradients: the modelled interpolar gradient in $\delta^{13}C$-$CH_4$ is reduced by ~75 % from ~0.4 ‰ to 0.1 ‰, and the gradient in $CH_4$ mixing ratios by ~22 % in the NO_WETLD scenario relative to the BASE scenario. Increasing high northern latitude wetland emissions by 50 % increases the interpolar difference in both $CH_4$ and $\delta^{13}C$-$CH_4$ in the INC_WETLD scenario relative to the BASE scenario. In this case, the gradient in $CH_4$ mole fractions is then slightly overestimated.

### 6.2.2 Varying the phase of the seasonal cycle

To investigate the impact of the prescribed phase of the seasonal cycle of high latitude wetland methane emissions on modelled atmospheric distributions of $CH_4$, $\delta^{13}C$-$CH_4$ and $\delta D$-$CH_4$, a further model scenario is run (DEL_WET) in which the seasonal cycle of this source is delayed by one month, resulting in a later spring kick-off in emissions and a decline in emissions that occurs later in autumn than in the BASE scenario. While this has a negligible influence on the modelled latitudinal gradient (not shown), delaying the high latitude wetland emission seasonal cycle by one month (so the summer emission season starts and finishes one month later in the year) has a notable impact on modelled seasonal variations in atmospheric methane and its isotopic composition (see Fig. 8). In this case, the model is better able to capture observed seasonal cycles in $CH_4$, $\delta^{13}C$-$CH_4$ and $\delta D$-$CH_4$.

These results do not support the existence of a large spring burst in wetland emissions as has been reported in other studies (e.g. Christensen et al., 2004; Song et al., 2012). To capture the correct timing of the $CH_4$ minima and $\delta^{13}C$-$CH_4$ and $\delta D$-$CH_4$ maxima, the model requires that there be no large contribution from wetland emissions until June, with peak emissions occurring between July and September (see Figure 1, Fung_Del scenario) Equally, to capture the correct timing of the summer/autumn increase in $CH_4$ mixing ratios and decrease in $\delta^{13}C$-$CH_4$, the model requires strong contributions from an isotopically light source continuing through to October. This could be from autumnal wetland emissions, as represented here. A large late-autumnal high northern latitude wetland source is supported by the recent work of Zona et al., (2016), who observed strong methane fluxes at an Arctic wetland site continuing well after the near-surface soil layer starts to freeze in late August or early September. Alternatively, it is possible that the comparison between modelled and observed $\delta^{13}C$-$CH_4$ (though not $CH_4$ mixing ratios) could be improved by prescribing a seasonal variation to the signature of high northern latitude wetland emissions as observed by Sriskantharajah et al. (2012).

Figure 1 shows that the seasonal cycle of the Fung_Del emissions used in the DEL_WET scenario is similar in phase to that generated by the LPJ-Bern model (Melton et al., 2013), with an emission peak occurring later in the year than other datasets. In a comparision of the FUNG and LPJ-Bern wetland emission datasets we found that the difference in emission seasonal cycles at 50-90°N is a consistent feature over these latitudes, rather than a result of differing geographical emission distributions between the two datasets. In an intercomparison of wetland methane emission models over West Siberia (WETCHIMP-WSL, Bohn et al., 2015), the late August peak in Siberian emissions in LPJ-Bern was found to be due to a late peak in wet mineral soil intensity, supplemented by a late peak in $CH_4$-producing area. The August peak in LPJ_Bern West Siberian emissions in WETCHIMP-WSL was in agreement with the Bousquet et al. (2011) atmospheric model inversion study.

### 6.3 Model sensitivity to the hydrate / thawing permafrost source

Methane emissions from ocean bottom decomposing hydrates and thawing permafrost in the Arctic are not well known due to uncertainties in the amount of carbon in permafrost, the sizes and locations of the methane hydrate deposits, the rate of

heat transfer through the ocean and sediments, and the fate of methane once it has been released into sea water (O'Connor et al. 2010). A recent study by Shakhova et al. (2014) estimated methane emissions of 17 Tg yr$^{-1}$ from the ESAS based on extrapolation of field observations in the Southern Laptev Sea. An emission of this magnitude represents a substantial reassessment of the high northern latitude methane budget, being equivalent to ~25 % of total estimated methane emissions above 50° N. A subsequent study by Berchet et al. (2016) reported that an ESAS flux of this magnitude was inconsistent with atmospheric observations, and used a statistical analysis of observations and model simulations to estimate an ESAS source of 0.5 to 4.3 Tg yr$^{-1}$.

To assess the sensitivity of the model to uncertainties in this high latitude methane source, we compare three scenarios in which methane emissions from the East Siberian Arctic Shelf region are assigned magnitudes of 0, 5 and 17 Tg yr$^{-1}$ (NO_HYD, BASE and INC_HYD scenarios respectively). These emissions are set to be constant throughout the year as about 10 % of the ESAS remains open water in winter due to the formation of polynyas, implying that it could be a source of $CH_4$ to the atmosphere year-round (Shakhova et al. 2015), and due to the lack of any further data on seasonality. However, it is possible that summer ESAS fluxes, when the region is ice-free, could be larger than winter fluxes (Berchet et al. 2016). The influence of these changes in emissions on the modelled latitudinal gradient is shown in Fig. 9. Although the magnitude of change in emission is small in comparison to the global budget (<~3 %), varying the strength of the ESAS source has a notable impact on modelled interpolar differences as the source is highly localised at high latitudes. In the scenario in which East Siberian Arctic Shelf emissions have been removed (NO_HYD), northern high latitude gradients in modelled $CH_4$ and $\delta^{13}C$-$CH_4$ are underestimated relative to observations. This demonstrates that the model does require a small, very high latitude, isotopically light source to capture observed latitudinal gradients, given the prescribed geographical distributions of emissions from other high latitude sources used in the BASE scenario. However, when ESAS hydrate emissions are increased to 17 Tg yr$^{-1}$ (INC_HYD), the model predicts a larger latitudinal gradient in $CH_4$ between mid and high northern latitudes than seen in the observations (Fig 9). This remains true when using modelled mixing ratios from measurement site locations, rather than a zonal mean (not shown). Therefore our model simulations do not support the existence of an East Siberian Arctic Shelf methane source of this magnitude, given the representations of other methane sources outlined in Table 1.

It is, however, possible that an East Siberian Arctic Shelf source of 17 Tg yr$^{-1}$ could be accommodated in our model set-up if adjustments were made to the representation of other high northern latitude sources within the model. At 30 Tg yr$^{-1}$, wetlands represent the largest single methane source in high latitude regions (Table 1), and therefore have the largest potential for flux adjustment. We consider an alternative scenario (WET_HYD) including emissions of 17 Tg yr$^{-1}$ from the East Siberian Arctic Shelf, but only 18 Tg yr$^{-1}$ from high northern latitude wetlands (i.e. high northern latitude wetland emissions are geographically uniformly reduced by 12 Tg yr$^{-1}$ and total NH emissions remain the same as in the BASE scenario). In this case, the modelled zonal mean latitudinal gradient of $CH_4$ is in better agreement with the observations than INC_HYD (Fig. 9), and modelled mixing ratios from measurement site locations have very close agreement with observations. However in WET_HYD, the zonal mean latitudinal gradient in $\delta^{13}C$-$CH_4$ is reduced relative to both observations and the BASE scenario

in northern mid-latitudes (Fig. 9). When WET_HYD modelled mixing ratios from measurement station locations are used rather than a zonal mean, this reduction in gradient is more apparent. This occurs as the isotopic signature of hydrate/permafrost emissions assigned in the model is larger than that of high latitude wetland emissions (~-55 ‰ compared to ~-70 ‰, see discussion below). In addition to the impact on the latitudinal gradient, the agreement of the model with

observed seasonal cycles of $CH_4$, $\delta^{13}C$-$CH_4$ and $\delta D$-$CH_4$ is also reduced in high northern latitudes following the 12 Tg yr$^{-1}$ reduction in high northern latitude wetland emissions (not shown). However, this is based on the use of constant ESAS emissions and inclusion of a seasonal cycle may influence our results. For example, if ESAS emissions with a $\delta^{13}C$ isotopic signature of -55 ‰ were assigned a seasonal cycle that peaked during the summer, along with wetland emissions, then this would likely lead to smaller differences in modelled seasonal cycles between WET_HYD and BASE.

These results are, at least partly, based on the assumption that the isotopic signatures assigned to high northern latitude wetlands and ocean floor hydrates/thawing permafrost are correct, and specifically that the $\delta^{13}C$ signature for wetland emissions is more negative than that for hydrates/permafrost. $\delta^{13}C$ signatures for Arctic wetland emissions have been determined in a number of studies and there is strong agreement that these emissions are highly depleted in $^{13}C$, with values <-65 ‰ (Fisher et al., 2011, Sriskantharajah et al., 2012, O'Shea et al., 2014). Our value of -70 ‰ is based on recent data

from the NERC MAMM (Methane in the Arctic: Measurements, process studies and Modelling) campaign (O'Shea et al., 2014). $\delta^{13}C$ signatures from ocean floor hydrates and permafrost are less well known and as far as we are aware, have not been published for the Laptev Sea region. Measurements taken from decomposing $CH_4$ hydrate in sediment cores in the Norwegian Arctic show a wide $\delta^{13}C$ isotopic range, from ~-72 ‰ to ~-46 ‰ (Milkov, 2005; Vaular et al., 2010, Fisher et al., 2011). However, methane released from the sea floor will be oxidised in the water column and enriched in $^{13}C$ before

reaching the atmosphere as methanotrophs in ocean water would preferentially consume the lighter isotope. Therefore the isotopic signature of emission to the atmosphere will be more enriched in $^{13}C$ (less negative $\delta^{13}C$) than the $\delta^{13}C$ values from sediment cores (Graves et al., 2015). A substantially lighter isotopic signature for ESAS methane emissions, as would be required to capture atmospheric $\delta^{13}C$-$CH_4$ observations, is possible, however it would require both (a) a very light initial isotopic composition on release at the sea floor and (b) very limited oxidation in the water column before release to the

atmosphere. These factors could be achieved with a shallow sea floor (as is present for the ESAS) and the formation of large methane bubbles.

To assess how a more negative $\delta^{13}C$ signature for ESAS hydrate/permafrost emissions would influence our model results, we construct a further scenario for $\delta^{13}C$-$CH_4$, WET_HYD_$\delta^{13}C$,  in which the ESAS source of 17 Tgyr$^{-1}$ is assigned a $\delta^{13}C$ signature of -70 ‰. In this case, the model simulates a much larger latitudinal gradient in $\delta^{13}C$-$CH_4$ in high northern latitudes

than is seen in the observations (Fig. 9).The agreement of the WET_HYD_$\delta^{13}C$ scenario with observed seasonal cycles of $CH_4$, $\delta^{13}C$-$CH_4$ and $\delta D$-$CH_4$ is reduced relative to BASE in high northern latitudes (not shown). However this is based on using constant aseasonal ESAS emissions in the model. If a seasonal cycle peaking during the summer was applied to ESAS emissions, it would likely become harder to distinguish between atmospheric $CH_4$ and $\delta^{13}C$-$CH_4$ seasonality due to ESAS

emissions and that due to high latitude wetland emissions in our study as both emission datasets would have similar seasonal cycles and $\delta^{13}$C isotopic compositions. Therefore, whether an ESAS source of 17 Tgyr$^{-1}$ can be accommodated in our global model along with a reduction in high northern latitude wetland emissions is highly dependent on the $\delta^{13}$C signature used for the respective sources, as well as potentially the seasonal cycle applied to the ESAS emissions. Our model simulations

indicate that if the ESAS source has a very negative $\delta^{13}$C signature (-70 ‰ or more negative), then such a large, localised, high latitude source would strongly influence global scale hemispheric gradients.

The sum of all other (mostly anthropogenic) sources >50° N is ~37 Tg yr$^{-1}$ (see Table 1). The isotopic compositions of these sources are all either similar to, or heavier than the isotopic signature assigned to the East Siberian Arctic Shelf source in our BASE scenario (-55 ‰). Therefore is it possible that East Siberian Arctic Shelf emissions of 17 Tg yr$^{-1}$ with a $\delta^{13}$C value of

-55 ‰ could be accommodated in model simulations of $CH_4$ and $\delta^{13}$C-$CH_4$, provided substantial reductions in high latitude anthropogenic emissions of methane (for example ~33 % across all sources) are also included in the simulations. In this case the agreement between the modelled and observed interpolar difference in $CH_4$ and $\delta^{13}$C-$CH_4$, and the high northern latitude seasonal cycles of $CH_4$, $\delta^{13}$C-$CH_4$ and $\delta$D-$CH_4$ could potentially be maintained. However, these scenarios could not be tested here as anthropogenic emissions were not tagged by latitude within the model. Emission totals for our BASE scenario and

the BASE scenario including a 33% reduction to anthropogenic emissions >50°N, both give anthropogenic emission totals within the  range of top-down and bottom-up emission estimates presented by Kirschke et al. (2013) (although towards the lower end for each source type when the 33% reduction is included).  Although within current ranges of uncertainty, such large flux adjustments to high latitude anthropogenic sources would indicate the presence of important errors in current inventories of high latitude emissions. In summary, to accommodate an ESAS source of ~17 Tg yr$^{-1}$ in our model

simulations requires a substantial revision of our emission scenario in high northern latitudes. We require either:

(a) A reduction in wetland emissions north of 50° N of ~40 % (i.e. totalling ~18 Tg yr$^{-1}$, a number just below the minimum of a range of process model studies), and ESAS emissions to have a seasonality and highly depleted isotopic signature similar to high northern latitude wetlands (i.e. peaking during the summer ice-free period).

(b) A reassessment of anthropogenic methane emission inventories in which total emissions above 50° N are reduced

by approximately 33 %, and ESAS emissions are emitted approximately constantly through the year with an isotopic signature close to anthropogenic emissions (~55 ‰)

(c) A combination of the above

(d) The inclusion of an additional, as yet unrepresented, high latitude sink, such as the boreal plant sink outlined in Sundqvist et al. (2012).

**7 Implications for Arctic sources**

Model studies disagree over the magnitude and seasonal distribution of high northern latitude wetland methane emissions (Melton et al., 2013; Bohn et al. 2015). This disagreement needs to be resolved in order to better predict future wetland

emissions in a warming climate. In this study, we find that high northern latitude wetland emissions have an important influence on both the magnitude and phase of high northern latitude seasonal cycles of $CH_4$ mixing ratios, $\delta^{13}$C-$CH_4$ and $\delta$D-$CH_4$. To date, measurements of $\delta$D source signatures are more limited than for $\delta^{13}$C, and uncertainties in source $\delta$D and $KIE^{H/D}$ values limit the conclusions that can be drawn from measurement-model comparisons of atmospheric data. However,

with improved data, our model study shows that atmospheric observations of $\delta$D-$CH_4$, as well as $\delta^{13}$C-$CH_4$ could provide an important constraint on current emissions from Arctic wetlands and inter-annual trends in this climate-sensitive source.

In our model simulations, the model's ability to capture the magnitude and observed seasonal cycles of $CH_4$ mixing ratios, $\delta^{13}$C-$CH_4$ and $\delta$D-$CH_4$ in high northern latitudes is much improved if the seasonal cycle of the Fung et al. (1991) wetland emissions is delayed by one month (i.e. the wetland emission season starts and finishes one month later than in the

prescribed dataset). As modelled atmospheric seasonal cycles at measurement station locations showed little sensitivity to emissions from north Asia (predominantly Siberia), this result is applicable to North American and North European wetland emissions. How this is interpreted will depend on the time-resolution of the emission dataset (one month for Fung et al., (1991)), and the temporal method of implementation in the model. In p-TOMCAT, emissions are linearly interpolated in time from the centre-point of the month. However, with improved temporal resolution of emissions, perhaps a better

agreement could be obtained without the need to delay the seasonal cycle.

Figure 1 shows a comparison of seasonal cycles in high northern latitude wetland emissions from Fung et al. (1991) compared to emission data from wetland process models obtained as part of the recent WETCHIMP model comparison (Melton et al. 2013) and the methane model inversion study of Bousquet et al. (2011). The resulting emission distribution from delaying the Fung et al. (1991) seasonal cycle by one month generally falls within the range of model uncertainties,

with the phase and shape of the seasonal cycle (though not emission magnitude) most closely matching that of the LPJ-Bern model.The delayed start to the emissions in FUNG_DEL results in notably smaller emissions in May than predicted by the other studies, excluding the atmospheric inversion study of Bousquet et al. (2011). The Bousquet et al. (2011) study obtained significant year to year differences in high latitude springtime emissions during the 1993 to 2009 time period considered. For their 1994-2004 period, emissions during May were significantly higher than for the years 2005 onwards, where they were

often negative (see Fig. 1). Low emissions in May could be a result of continued snow cover at high latitudes or high water levels during the melt season limiting the amount of $CH_4$ released to the atmosphere due to oxidation in the water column. In addition, spring increases in $CH_4$ uptake by oxic forest soils and/or the canopy could contribute towards lower net emissions from high latitudes in May (Sundqvist et al. 2012). p-TOMCAT also requires a larger autumnal isotopically 'light' methane source than predicted by most wetland models to capture observed seasonal cycles of $CH_4$, $\delta^{13}$C-$CH_4$ and $\delta$D-$CH_4$. This

result is consistent with a recent study by Zona et al., (2016) measuring year-round wetland fluxes at an Arctic wetland site. They found large methane fluxes continuing throughout the 'zero curtain' period, where subsurface soil temperatures remain active at ~0°C before freezing around December time, partly due to the insulating effects of snow cover. Other possible contributions towards an additional, isotopically light, autumnal methane source include processes releasing methane during tundra freezing (Mastepanov et al. 2008).

Using current literature estimates for northern high latitude methane emissions, our study suggests an ESAS methane source in the lower half of published estimated ranges (0.5 to 17 Tg yr$^{-1}$). This is in agreement with the study by Berchet et al. (2016), which used synoptic data from long-term methane measurement sites to constrain ESAS emissions from 0.5 to 4.3 Tg yr$^{-1}$. We find that substantial adjustments in estimates of high latitude methane source flux magnitudes or isotopic source

signatures are required in order to reconcile East Siberian Arctic Shelf emissions as large as 17 Tg yr$^{-1}$ with global scale atmospheric observations of $CH_4$ and $\delta^{13}C$-$CH_4$. Depending on currently-lacking information on the seasonality and isotopic signature of an ESAS source, these include reducing high northern latitude wetland emissions by ~40 % (to a value just below the minimum of a range of values predicted by process models), reducing high northern latitude emissions from anthropogenic emission inventories by ~33 % or a combination of the two. Alternatively, a missing seasonal sink, such as

the destruction of methane by boreal vegetation suggested by Sundqvist et al. (2012) could help reconcile large emissions from the ESAS with global scale atmospheric observations. Further information on the isotopic signature and seasonality of an ESAS source would be of benefit in distinguishing between possible scenarios.

**Acknowledgements**

The authors acknowledge funding from the NERC MAMM project (NE/I029161/1 and NE/I028874/1). NJW and JAP thank

NCAS-Climate for funding. NJW and JAP also thank NERC for funding via the projects NE/K004964/1 and NE/I010750/1. DL, JF, REF and EGN thank NERC for funding via projects NE/K006045/1 and NE/I014683/1. This study was also supported by the ERC under the ACCI project, grant number 267760.

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

Table 1. Global methane source magnitudes and isotopic signatures used in p-TOMCAT

| Surface Source/Sink | Global flux (Tg/yr) | High latitude (>50°N) flux (Tg/yr) | $\delta^{13}$C-CH$_4$ (‰) | $\delta$D-CH$_4$ (‰) |
|---|---|---|---|---|
| Northern Wetlands | 30[1] | 30.0 | -70[d,h,l,n*] | -360[f,n*] |
| Tropical Wetlands | 200[1] | 0.0 | -55[b,m*] | -320[g,o*] |
| Hydrates | 5[1] | 5.0 | -55[d*] | -190[p] |
| Coal | 40[2,3] | 3.2 | -50[i,q*] | -140[p] |
| Gas | 63[2,3] | 15.3 | -40[b,n*] | -185[i,j,n*] |
| Biomass burning | 31[4] | 3.1 | -26[b] | -210[k] |
| Ruminants | 110[2] | 8.0 | -63[a,c*] | -360[a*] |
| Landfills | 27[2] | 4.6 | -53[b] | -310[i,p] |
| Sewage | 29[2] | 1.8 | -57[b] | -310[r] |
| Rice | 33[2] | 0.0 | -62[b,g,m*] | -330[p*] |
| Termites | 20[1] | 1.1 | -57[e,m*] | -390[p] |
| Total | 588 | 72.1 | | |

The geographical and seasonal distribution of methane flux data is based on [1]Fung et al., 1991, [2]EDGAR v4.1 (http://edgar.jrc.ec.europa.eu/overview.php?v=41) for 2005, [3]Gurney et al., 2005, and [4]Van der Werf et al., 2006. Source isotopic signature data are based on reported values from: [a]Bilek et al., 2001, [b]Dlugokencky et al., 2011, [c]Levin et al., 1993, [d]Fisher et al., 2011, [e]Gupta et al., 1996, [f]Nakagawa et al., 2002a, [g]Nakagawa et al., 2002b, [h]O'Shea et al., 2014, [i]Quay et al., 1999, [j]Schoell, 1980, [k]Snover et al., 2000, [l]Sriskantharajah et al., 2012, [m]Tyler et al., 1988, [n]Umezawa et al., 2012, [o]Waldron et al., 1999, [p]Whiticar and Schaefer, 2007 , [q]Zazzeri et al., 2015, [r]value used taken from landfill data, *value is within a range of quoted literature estimates.

Table 2.

| Scenario | Difference from BASE Scenario |
| --- | --- |
| BASE | - |
| DEC_KIE | $KIE^{(CH4+OH)}/KIE^{(CH3D+OH)}$ is decreased from 1.29 to 1.16[a] |
| WETLD_δD | δD signature for wetland emissions >50°N changed to -500‰ |
| NO_WETLD | Wetland emissions >50°N removed |
| INC_WETLD | Wetland emissions >50°N increased by 50% to 45 Tg/yr |
| DEL_WET | Seasonal cycle of wetland emissions >50°N delayed by one month throughout the year |
| NO_HYD | Hydrate emissions removed |
| INC_HYD | Hydrate emissions increased to 17 Tg/yr |
| WET_HYD | Hydrate emissions increased to 17 Tg/yr and wetland emissions decreased to 18 Tg/yr |
| WET_HYD_δ13C | As WET_HYD, except isotopic signature for ESAS emissions is changed to -70 ‰ |

[a]See Table S1.

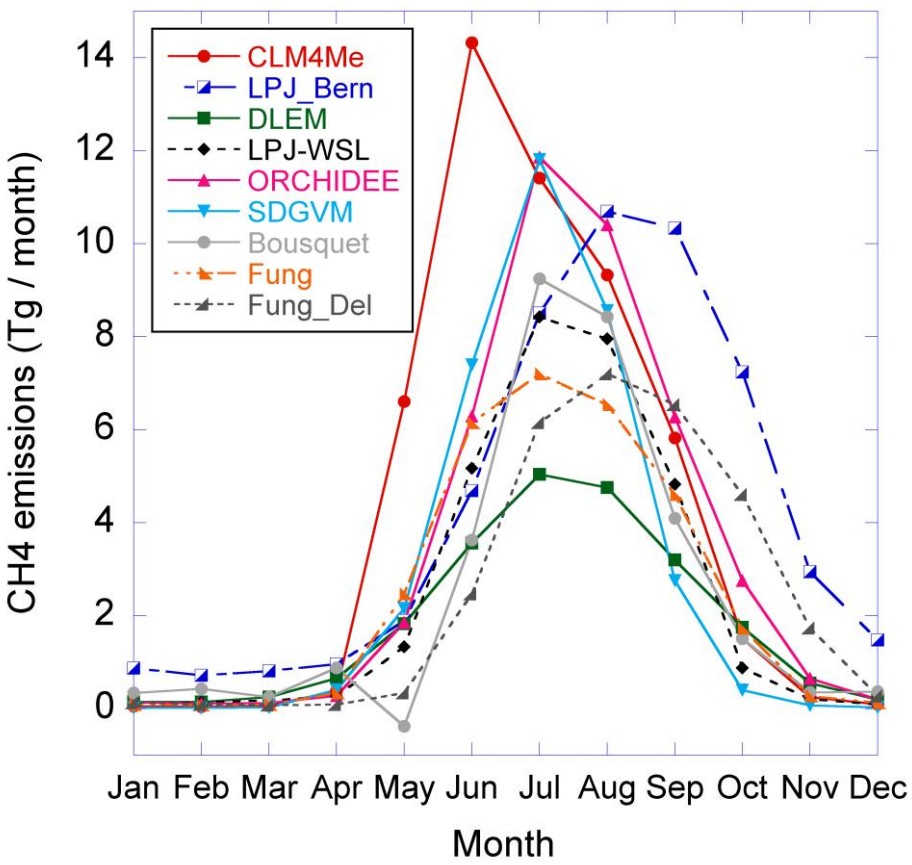

**Figure 1: A comparison of seasonal cycles in northern wetland emissions (>50° N) from Fung et al. (1991) (Fung), Fung et al. (1991) with a seasonal cycle delayed by one month (Fung_Del), mean annual emission data for 1993-2004 from wetland process models obtained as part of the recent WETCHIMP model comparison (CLM4Me, LPJ_Bern, DLEM, WSL, ORCHIDEE, SDGVM; Melton et al., 2013) and mean annual emission data for -2005-2009 from the methane model inversion study of Bousquet et al., (2011).**

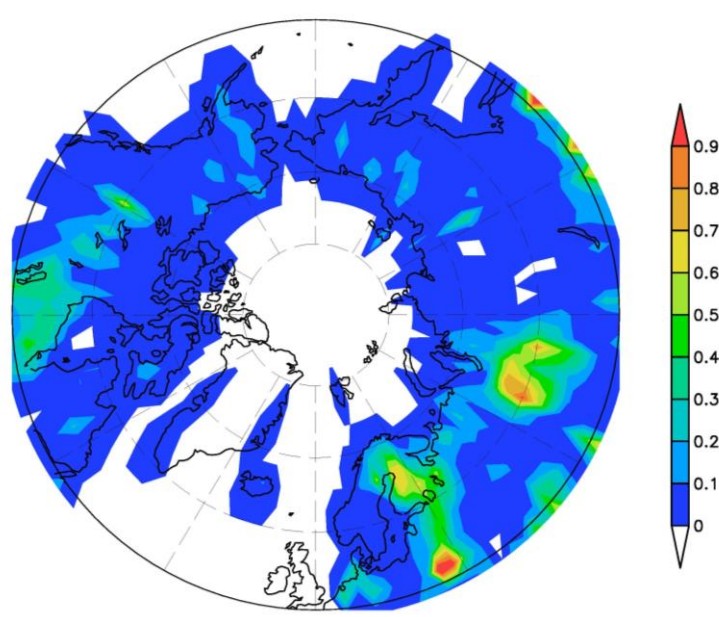

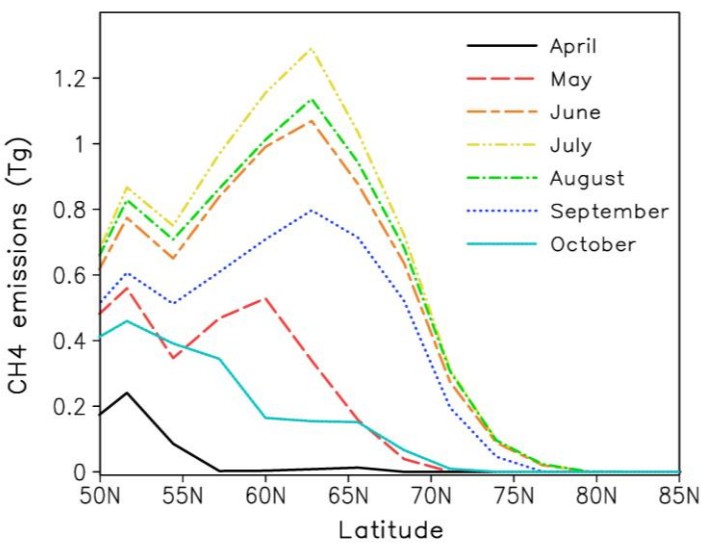

Figure 2: a) The geographical distribution of annual mean wetland emissions (mg/m$^2$/hr) above 50º N used in the model simulations. b) Zonally summed monthly CH$_4$ emissions from a) for April to September. Emissions in a) and b) have been interpolated to the model resolution (~2.8° x 2.8°) and are based on Fung et al., 1991.

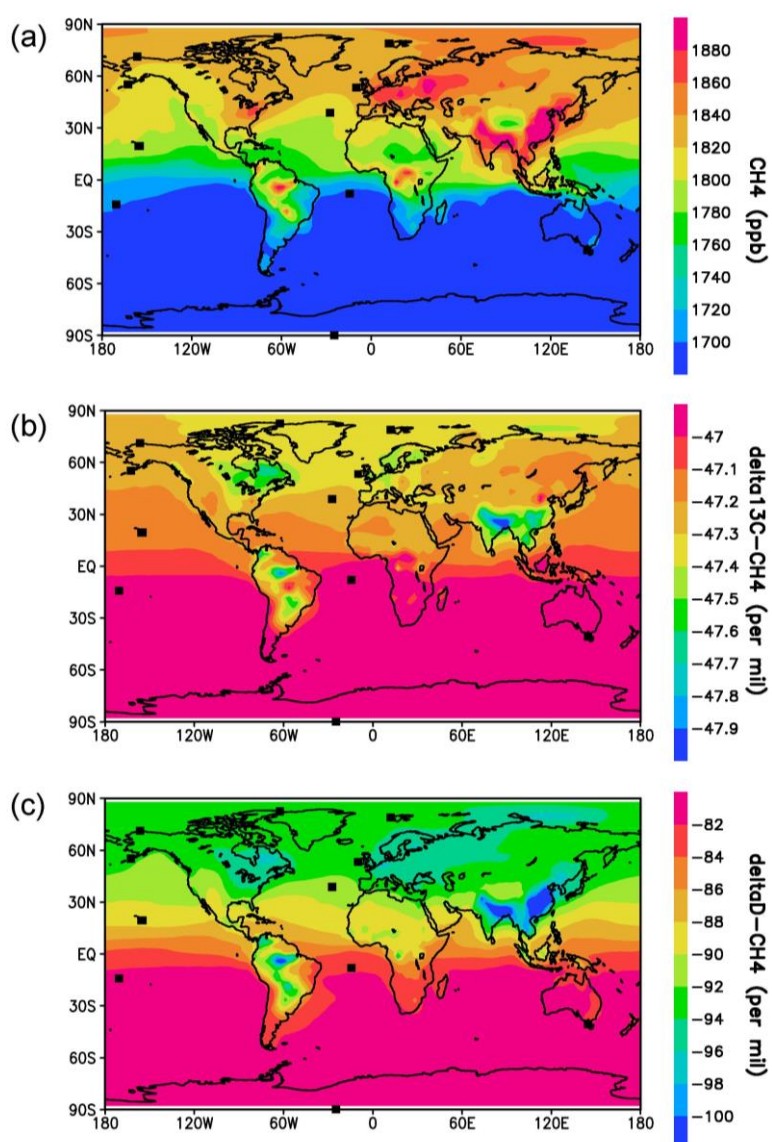

**Figure 3: Modelled global annual mean surface distributions of (a) CH₄, (b) δ¹³C-CH₄ and (c)δD-CH₄ for the p-TOMCAT BASE scenario. Locations of measurement data sites used in this study are marked as black squares.**

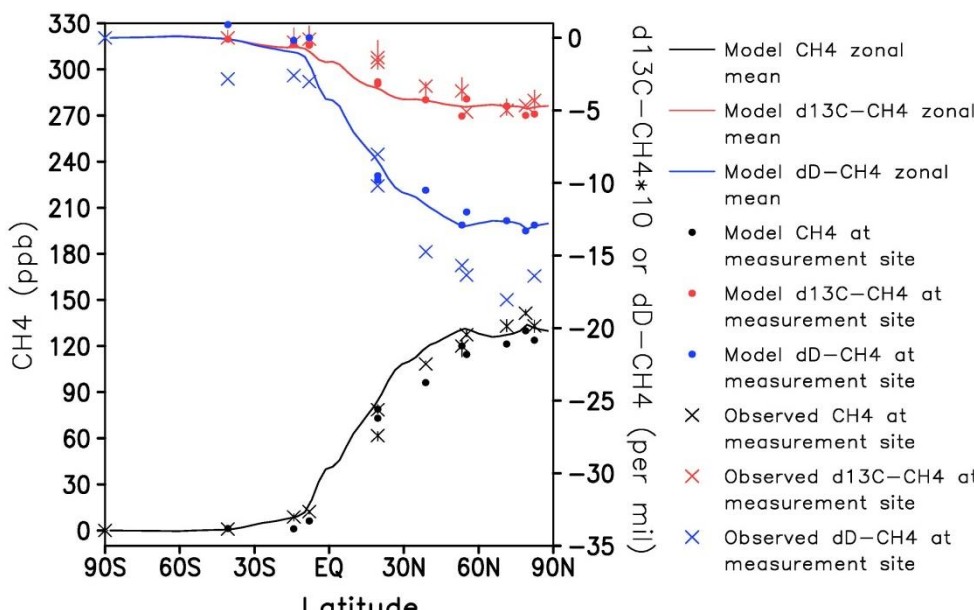

Figure 4: The difference between surface annual mean CH$_4$, δ$^{13}$C-CH$_4$ and δD-CH$_4$ and South Pole annual mean values for CH$_4$, δ$^{13}$C-CH$_4$ and δD-CH$_4$. Results from the p-TOMCAT BASE scenario (including sampling the model at station locations) are compared to NOAA-ESRL and CU-INSTAAR observations. Where there are sufficient data available in the 2005 to 2009 period, the range in annual mean station-South Pole observed differences is represented by a vertical bar. CH$_4$ mixing ratios are shown in black, δ$^{13}$C-CH$_4$ in red and δD-CH$_4$ in blue. Variations in δ$^{13}$C-CH$_4$ have been multiplied by a factor of 10.

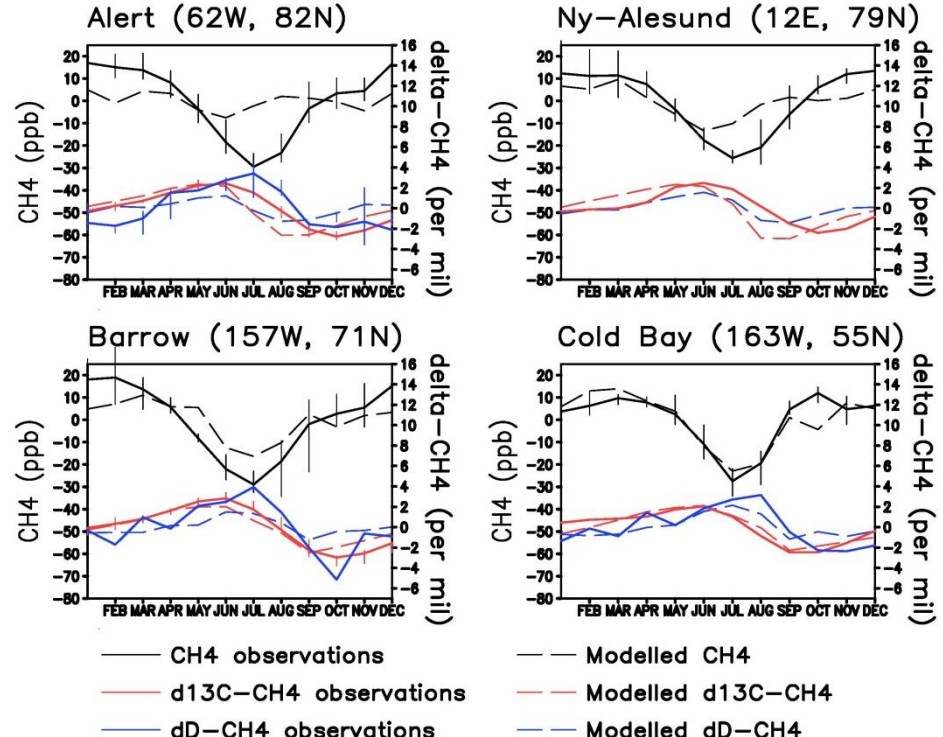

**Figure 5: A comparison of modelled seasonal cycles of $CH_4$, $\delta^{13}C$-$CH_4$ and $\delta D$-$CH_4$ from the p-TOMCAT BASE scenario and NOAA-ESRL and CU-INSTAAR observations ($\delta D$-$CH_4$ is not shown for Ny-Alesund due to insufficient data). Annual means have been subtracted from both the model and measurement data. Variations in $\delta^{13}C$-$CH_4$ have been multiplied by a factor of 10. Where there are sufficient data available in the 2005 to 2009 period, the range of observed monthly mean values relative to the annual mean is represented by a vertical bar.**

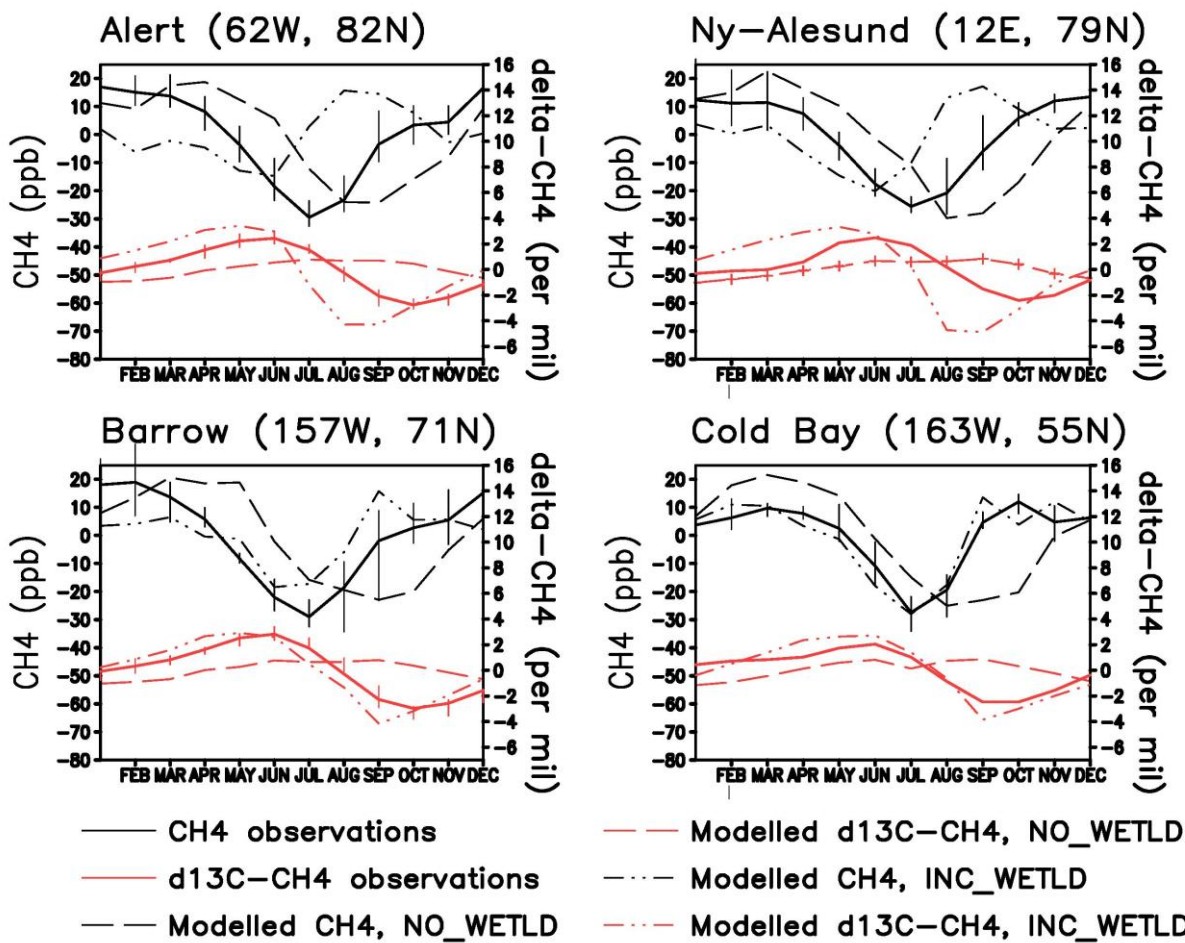

**Figure 6: Modelled seasonal cycles of CH₄ and δ¹³C-CH₄ compared to NOAA-ESRL and CU-INSTAAR observations. Annual means have been subtracted from both the model and measurement data. Black represents CH₄ mole fractions and red represents δ¹³C-CH₄. Where there are sufficient data available in the 2005 to 2009 period, the range of observed monthly mean values relative to the annual mean is represented by a vertical bar. Dashed lines represent model results from the NO_WETLD scenario (where wetland emissions >50° N have been removed relative to BASE). Dot-dot-dash lines represent model results from the INC_WETLD scenario (where wetland emissions >50° N have been increased by 50 % relative to BASE). Variations in δ¹³C-CH₄ have been multiplied by a factor of 10.**

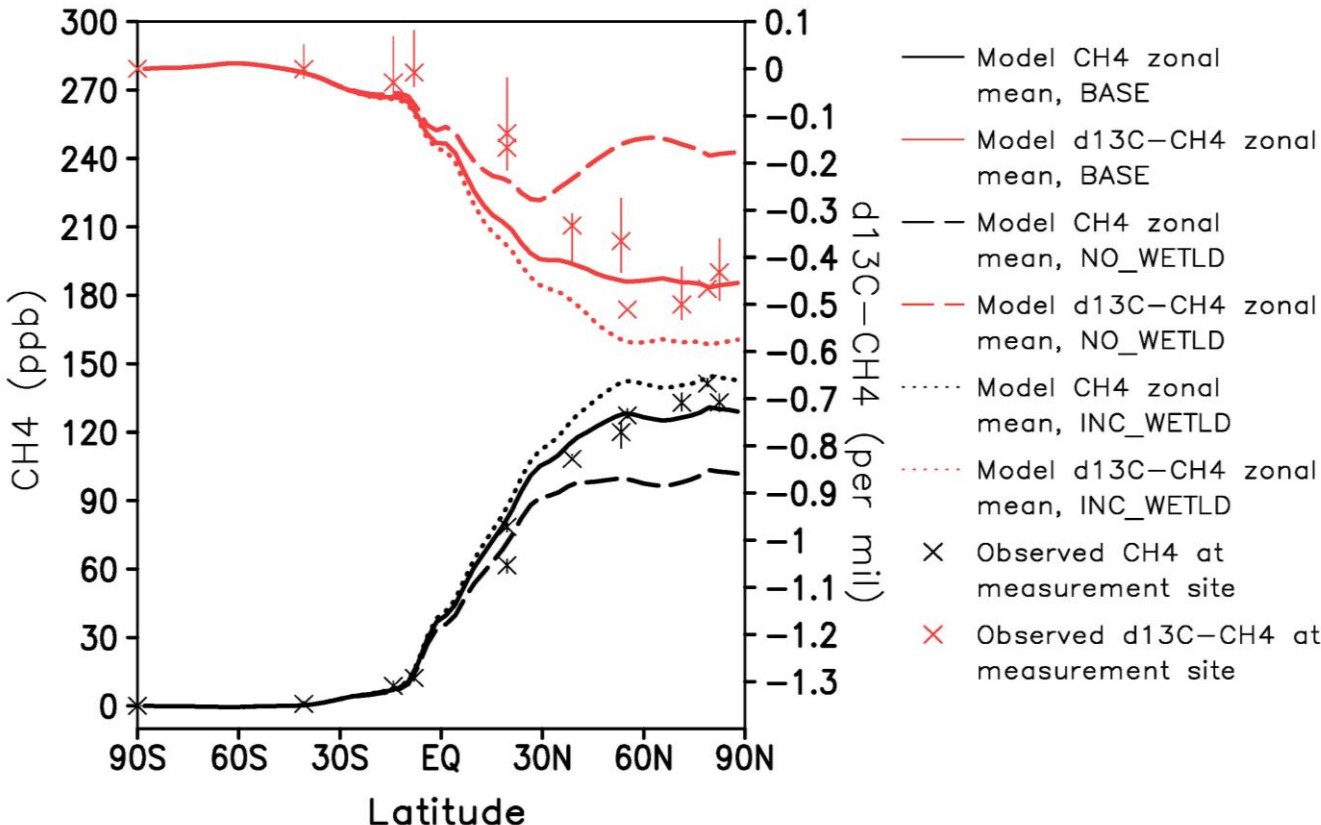

**Figure 7: The difference between surface annual mean CH$_4$ and δ$^{13}$C-CH$_4$ and South Pole annual mean values. Model results are compared to NOAA-ESRL and CU-INSTAAR observations. Black represents CH$_4$ mixing ratios and red, δ$^{13}$C-CH$_4$ fractionations. Where there are sufficient data available in the 2005 to 2009 period, the range in annual mean station-South Pole observed differences is represented a vertical bar. Solid lines represent model results from the BASE scenario. Dashed lines represent model results from the NO_WET scenario (where wetland emissions >50° N have been removed relative to BASE). Dotted lines represent model results from the INC_WETLD scenario (where wetland emissions >50° N have been increased by 50 % relative to BASE).**

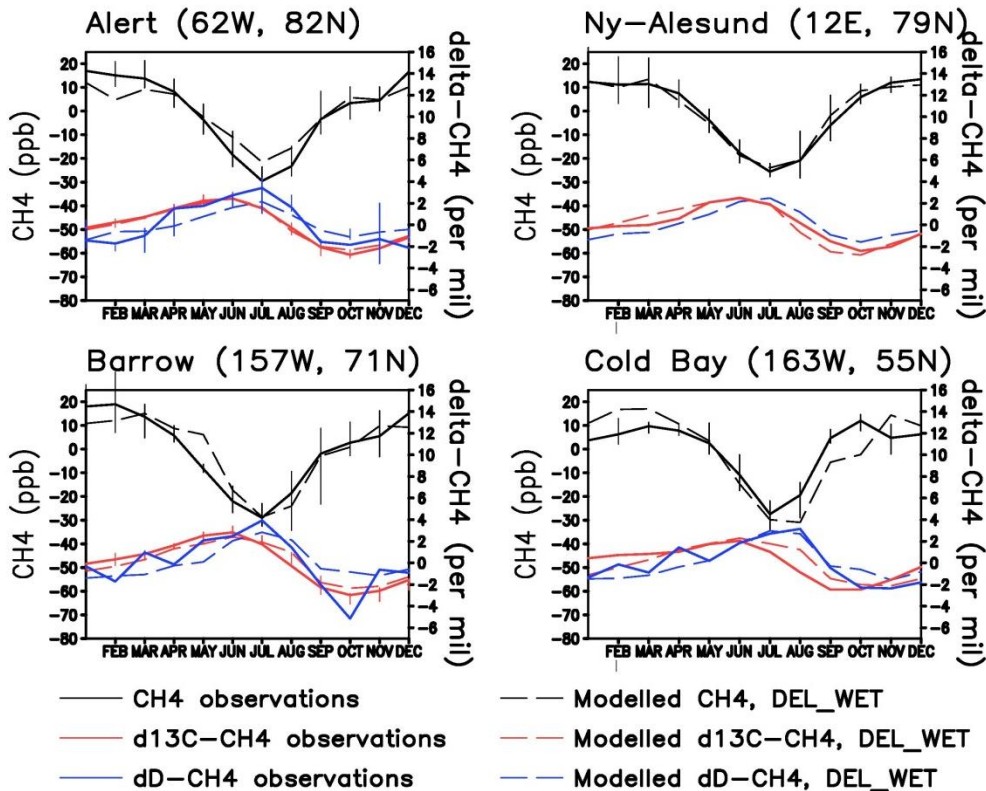

**Figure 8: As Figure 5, except showing model results from the DEL_WET scenario (where the seasonal cycle of wetland methane emissions from >50° N has been delayed by one month relative to BASE). Variations in δ$^{13}$C-CH$_4$ have been multiplied by a factor of 10.**

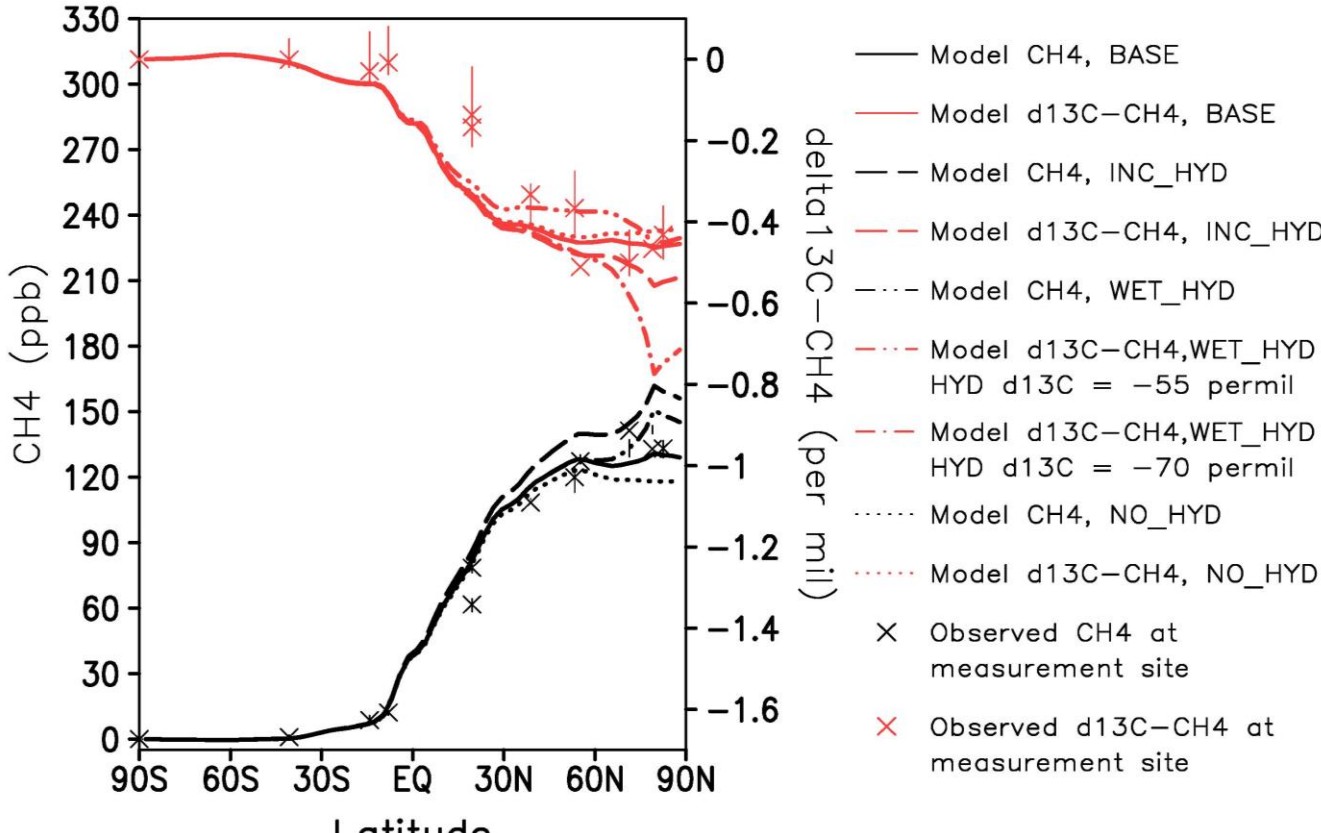

**Figure 9: The difference between surface annual mean modelled latitudinal gradients in CH$_4$ and δ$^{13}$C-CH$_4$ and South Pole annual mean values. Model results are compared to NOAA-ESRL and CU-INSTAAR observations. Black represents CH$_4$ mixing ratios and red, δ$^{13}$C-CH$_4$ fractionations. Where there are sufficient data available in the 2005 to 2009 period, the range in annual mean station-South Pole observed differences is represented a vertical bar. Solid lines represent model results from the BASE emission scenario. Dashed lines represent model results from the INC_HYD scenario (where hydrate emissions have been increased by 12 Tg yr$^{-1}$ to 17 Tg yr$^{-1}$ relative to BASE). Dotted lines represent model results from the NO_HYD scenario (where emissions from methane hydrates are removed relative to BASE). Dot-dash lines show model results from the WET_HYD scenario (where hydrate emissions are increased by 12 Tg yr$^{-1}$ to 17 Tg yr$^{-1}$ and wetland emissions > 50° N are reduced by 12 Tg yr$^{-1}$ relative to BASE). Dot-dot-dash lines represent emission magnitudes as for the WET_HYD scenario, but with an isotopic fractionation for hydrate emissions of -70‰.**