# Peer review of "Using $\delta^{13}$ C-CH4 and $\delta$ D-CH4 to constrain Arctic methane emissions"

_Atmospheric Chemistry and Physics, 2016_

## Referee Comment (RC1) · Anonymous Referee #1 · 25 Jul 2016

Overview:

The manuscript "Using $\delta^{13}$C-CH$_4$ and $\delta$D-CH$_4$ to constrain Arctic methane emissions" by Warwick et al. describes the results of a modelling study of Arctic wetland and hydrate emissions, in which the simulated concentrations of CH4, along with the associated $\delta^{13}$C-CH$_4$ and $\delta$D-CH$_4$ ratios, are compared to observations made at a number of high-latitude Northern Hemisphere measurement sites. The latitudinal gradient of these isotopologues is also assessed in comparison to observations. Finally, in an attempt to improve our current understanding of methane emissions from the Arctic, the effect of changes made to the wetland and hydrates emission inventories in the region is investigated.

Overall the manuscript is very well written, with few technical corrections necessary. The figures are generally quite clear and well chosen, although some small alterations are necessary for a couple of them. The methods used in this manuscript provide a neat way of assessing the accuracy of some of the current methane inventories used in atmospheric models, and the improvement in the comparisons with observations after the seasonal cycle of the wetland emissions is altered is striking. Using the three isotope ratios of methane as a 'triple check' on the seasonal cycle of the emissions works well and provides extra clues as to the timing and magnitude of emissions in the region. Finally the examination of the magnitude of hydrate emissions in the Arctic, whilst brief, does indicate that some recent estimates of emissions from this source may be too large.

My main reservation is that the conclusions drawn are dependent on a single (fairly old) wetland inventory, and there is no discussion on the impact that this fact might have on results. Is the relative geographical distribution of wetland emissions important for your conclusions to be substantiated? See general comments for more details.

I recommend this manuscript for publication after these revisions have been carried out.

Comments:

Page 1, lines 20-29: These paragraphs could use some extra references. You describe the recent changes in the methane growth rate without referring to any sources for this information ('2007… rapid methane increase','growth was strongest in the tropics', etc.), and there is also no reference for the

assertion that fossil fuel changes could play a role in the global growth rate or that Arctic emissions are poorly quantified.

Page 5, lines 5-9: In this work, you have used observations averaged over 2005-2009 and model meteorology for 2009 only, but in order to show that the OH fields used in the study are to some extent accurate, you show comparisons with MCF concentrations at one site for the year 2011. To be consistent with the meteorology used for the later figures, can you show 2009 concentrations here? MCF measurements should also be available at Alert, Canada. Does the model also capture the seasonal cycle that far north?

Page 5, line 16:  My main reservation with this study is related to the emissions inventories used. The Fung wetland inventory is now 25 years old, and whilst it generally does a good job, I think it is worth at least discussing the idea that the distribution of emissions in this inventory may not be correct. Since all of your observation sites are located in the US and Europe, are the observed seasonal cycles sensitive to the significant emissions from Siberia, or is the cycle only of the local emissions important?
Ideally, you'd carry out a supplementary model-run in which an alternative wetland scenario is used. The Bousquet (2011) inversion inventory, for example, assimilated observations of $CH_4$ made throughout the Arctic, and would likely, therefore, be able to capture the seasonal cycle of Arctic $CH_4$ well. However, according to Figure 1, it does not show the same delayed seasonal cycle and large magnitude of autumnal emissions required in your FUNG_DEL cycle in order to capture the seasonal cycle of $CH_4$. Also, as far I can tell, it has not been compared to observations of methane isotopologues before, and doing so may back up your conclusions that significant emissions deeper into the autumn are necessary.
Related to this, I note that you used the GFEDv2 biomass burning inventory. Version 4 of this inventory is now available, and any changes to the impact that the heavier $\delta^{13}C$-$CH_4$ has at these locations might affect your conclusions. However, I accept that the relative contribution of biomass burning emissions compared to wetland emissions at these latitudes is probably very small and therefore unlikely to have an effect unless emissions are local to the measurements.
If further simulations are not possible, I think a discussion of the effect of your choices on your results should be included in the results section.

Page 5, line 16: This is the first mention of the BASE scenario. You should make explicit that here, 'BASE' refers to the control experiment that uses the emissions described in the previous section, rather than some model set-up.

Page 6, line 6-10: This paragraph needs a little more detail. You have not previously described the locations of those measurements made further south than Cold Bay (perhaps they could also be included in Figure 4?). You say that the gradient in $\delta$D-CH$_4$ is captured, and also that it is underestimated in the NH mid latitudes. Can you explain more clearly? It looks to me that perhaps the $\delta$D-CH$_4$ is mostly captured quite well as far to 50S, but that using the South Pole value as a baseline is shifting the model away from the observations. Perhaps it's the SH gradient that isn't captured, rather than the NH gradient?

Page 8, line 12: It's a shame that there are no $\delta$D-CH$_4$ ratios included here for completeness, but since the changes to the wetland emissions in this section of the study don't improve simulated CH$_4$ or $\delta$C$^{13}$-CH$_4$ concentrations, I understand the reluctance to carry out the runs.

Page 8, line 18: The name "WETLD_X2" is a little misleading, as emissions have been increased only by 50%. Can you change this name?

Page 11, line 15: Are the model lines here full zonal means across all longitudes? If so, is there any impact on the comparisons at the sites in the Arctic if you compare only at the measurement locations? I think the plot would be too busy if you included these comparisons within it, but you could mention it in the text if there is any effect.

Figure 3: I think that this plot could be a little clearer. Can you include the locations of the measurement sites here (or in Figure 4)? Is this an annual mean or is it the peak summertime emissions? Can you differentiate between regions where wetland emissions are zero and where they're just smaller than the lowest value in your colourbar?
Perhaps you could include a similar second panel showing the standard deviation of the emissions, or the month during which emissions peak (or at least mention it in the text)? i.e. do emissions peak in July everywhere in the Artic, or does it vary by region?

Figure 10: Can you differentiate the lines more  clearly in this plot? The difference between the dash, dot, dot-dash and dot-dot-dash lines is not obvious enough in a plot of this size (especially as they only deviate in a small subsection of latitudes).

Technical corrections:

Page 1, line 11 and throughout: I find the use of the term 'coloured' throughout the manuscript to describe the different tracers a bit odd, although I accept that it can be a difficult idea to describe well. I'd suggest changing to the term 'tagged' or similar for clarity.

Page 2, line 3: "to-date" -> "to date" (no hyphen)

Page 12, line 29: "May-time emissions" ->  "May emissions"/"emissions than predicted in May"

Page 13, line 13: "currently lacking" -> "currently-lacking"

---

## Referee Comment (RC2) · Anonymous Referee #2 · 23 Aug 2016

General

The manuscript 'Using d13C-CH4 and dD-CH4 to constrain Arctic methane emissions' by Warwick and co-workers presents a study of atmospheric CH4 and its isotopes (d13C and dD) in the Arctic. Model simulations of all three components together with observations are used in a qualitative way to draw conclusions on the main arctic emission sources and their seasonal behavior. The analysis applies state-of-the-art modeling techniques and the methods and results are generally presented with a clear language and structure. The work clearly adds an important piece of information entangling the contributions of different methane sources and will probably help to further improve process models that are of paramount importance to understand future climate-emission feedbacks in the Arctic. I only have a few minor comments that can be addressed in the revised text but will, most likely, not require any major changes in

the analysis.

Major comments

One possibly more important remark has to do with the chosen wetland emissions in the presented simulations. Why was the rather old Feng et al. dataset chosen as a reference? As can be seen in Figure 1 the LPJ-Bern emissions already follow the suggested delay in summer-time emissions. Related to this: What is the influence of the spatial distribution of the chosen wetland emissions. Could the suggested shift in emissions also stem from an erroneous distribution in space rather than in time? How different is the spatial distribution of LPJ-Bern as compared to Feng et al?

Minor comments

P1, l18/19: Clarify if by inventories you are referring to purely anthropogenic emissions here.

P1, l26: What is the status of the Nisbet et al. publication? If not yet published another reference is needed here.

P2, l33: 'In this study': Does this still refer to Berchet et al. or the current study?

P3, l6/7: The given reference is rather old. Please give some newer references and a total amount of emissions here. See for example Kirschke et al. 2013 for some numbers and additional references.

P3, l8: The additional stratospheric sink by Cl and O1D should be mentioned here as well, although it probably adds little to the seasonality.

P3, l22-24: You should mention the work by Rigby et al. here as well, who already ran a CH4 isotope model (both d13C and dD) to evaluate the benefits of atmospheric isotopic observations: Rigby et al. 2012, JGR, VOL. 117, D12312, doi:10.1029/2011JD017384.

P3, l30f: In figures 5, 8 and 10 more than these 4 sites are used for comparison. Please mention which other sites are used in the figures.

P4, l16-18: For which period are these values given?

P5, l2: Which functional relationship was actually used to calculate the OH reaction rate coefficient? Reference or equation.

P5, l9f: To put the importance of these other reactions into perspective, could you give an average lifetime of CH4 wrt to stratospheric and MBL reactions as well?

P5, Table S2: Table S2 should be integrated in the main paper, since it is an essential information for the study. However, in the table it should be clarified which source is taken from which reference and how seasonality is considered.

P5, l24f: Which influence may the spin-up still have on the results. The spin-up basically leads to an initial state in which sources and sinks are in equilibrium. Is this adequate for the study period or could it be important to start from a background that is not in perfect equilibrium (due to steadily increasing emissions in reality).

P6, l22: 'with the seasonal cycle'. Does this refer to the observed seasonal cycle?

P9, l2f: Fig. 1 should be mentioned here again, which shows the emission profiles of Fung and Fung delayed.

P9, l6: 'forward': To me this is confusing. I would call it shifting the seasonal cycle backward.

p10, l1-3: Berchet et al. 2016 clearly showed a strong seasonal cycle for emissions from ESAS, with a summertime peak in the order of what was suggested by Shakhova et al. 2012 for the whole year (see Fig. 5 in Berchet et al. 2016), i.e. 10-15 Tg/yr. In contrast, Berchet et al. suggest close to zero emissions in winter. How could this seasonality, that was not considered in the current model analysis, change the drawn conclusions concerning the impossibility to accommodate the ESAS flux as suggested by Shakhova et al.?

Section 6.2: Please comment on the good agreement of the delayed temporal development used in your sensitivity run and that simulated by LPJ-Bern (as seen in Fig. 1). Does their model version already include possible CH4 emissions after freezing of the top soil?

P11, l21-23: This thought should be given some more discussion. Would it still be possible to accommodate a 50 % reduction in high-latitude anthropogenic emissions (as compared with the BASE run) within the range given by previous studies (e.g. Kirschke et al.).

Figure 2: Should be part of the supplement. It is not essential to the study, but 'only' demonstrates that the applied model seems to perform reasonably well in terms of OH degradation.

Figures 6 and 9: Why is there no observed dD for Ny-Alesund? If it was not observed there, it should be mentioned somewhere in the text.

Adding a table summarizing all model runs (BASE and sensitivities) and their specific settings would be nice. All the sensitivities should be given an abbreviation/name (not done in all cases) in this table and in the text so that it is easier to quickly identify what the specifics of a certain run are.

---

## Author Comment (AC1) · 21 Oct 2016

***Reply to Referee #1 (bold italics).***

Overview:

The manuscript "Using δ1 C-CH4 and δD-CH4 to constrain Arctic methane emissions" by Warwick et al. describes the results of a modelling study of Arctic wetland and hydrate emissions, in which the simulated concentrations of CH4, along with the associated δ13C-CH4 and δD-CH4 ratios, are compared to observations made at a number of high-latitude Northern Hemisphere measurement sites. The latitudinal gradient of these isotopologues is also assessed in comparison to observations. Finally, in an attempt to improve our current understanding of methane emissions from the Arctic, the effect of changes made to the wetland and hydrates emission inventories in the region is investigated.

Overall the manuscript is very well written, with few technical corrections necessary. The figures are generally quite clear and well chosen, although some small alterations are necessary for a couple of them. The methods used in this manuscript provide a neat way of assessing the accuracy of some of the current methane inventories used in atmospheric models, and the improvement in the comparisons with observations after the seasonal cycle of the wetland emissions is altered is striking. Using the three isotope ratios of methane as a 'triple check' on the seasonal cycle of the emissions works well and provides extra clues as to the timing and magnitude of emissions in the region. Finally the examination of the magnitude of hydrate emissions in the Arctic, whilst brief, does indicate that some recent estimates of emissions from this source may be too large.

My main reservation is that the conclusions drawn are dependent on a single (fairly old) wetland inventory, and there is no discussion on the impact that this fact might have on results. Is the relative geographical distribution of wetland emissions important for your conclusions to be substantiated? See general comments for more details.

I recommend this manuscript for publication after these revisions have been carried out.

***We thank the referee for their very helpful comments and suggestions which have helped improve the manuscript.***

***Our response to the comment on our use of the Fung et al., 1991 emission inventory is included in the replies to the general comments below.***

Comments:

1. Page 1, lines 20-29: These paragraphs could use some extra references. You describe the recent changes in the methane growth rate without referring to any sources for this information ('2007... rapid methane increase','growth was strongest in the tropics', etc.), and there is also no reference for the assertion

that fossil fuel changes could play a role in the global growth rate or that Arctic emissions are poorly quantified.

***Several references have now been added to this paragraph (see P1, l20-28), including those for the role of fossil fuel changes. The poor quantification of Arctic emissions is discussed (and referenced) by source type in the following paragraphs of the introduction.***

2. Page 5, lines 5-9: In this work, you have used observations averaged over 2005-2009 and model meteorology for 2009 only, but in order to show that the OH fields used in the study are to some extent accurate, you show comparisons with MCF concentrations at one site for the year 2011. To be consistent with the meteorology used for the later figures, can you show 2009 concentrations here? MCF measurements should also be available at Alert, Canada. Does the model also capture the seasonal cycle that far north?

***The figure has been changed to include observed MCF mixing ratios for 2009. As for the 2011 data, there is good agreement between modelled and observed seasonal cycles. As requested by Referee #2, we have now moved this figure into the supplementary information (Fig. S1).***

***MCF measurements are also available at Alert, however this data contains significantly more noise and lacks a smooth seasonal cycle. The model predicts a smooth MCF seasonal cycle at Alert, similar to that modelled at Barrow, but with a slightly smaller amplitude. The modelled MCF seasonal cycle at Alert compares well to the observed Alert seasonal cycle, minus the noise (i.e. capturing the approx. timing of peak/minimum mixing ratios). However, due to the additional noise in the Alert observations we have chosen to present only the Barrow data.***

3. Page 5, line 16: My main reservation with this study is related to the emissions inventories used. The Fung wetland inventory is now 25 years old, and whilst it generally does a good job, I think it is worth at least discussing the idea that the distribution of emissions in this inventory may not be correct. Since all of your observation sites are located in the US and Europe, are the observed seasonal cycles sensitive to the significant emissions from Siberia, or is the cycle only of the local emissions important?

***In our model simulations, high latitude northern wetland emissions from Asia, Europe and America were coloured or 'tagged' separately. Our results show us that modelled seasonal cycles at presented measurement sites are predominantly influenced by high northern latitude wetland emissions from America and/or Europe, with little sensitivity to Siberian emissions. We found that altering the Fung***

*emission distribution in a simple way via varying the relative emission strengths associated with these regional tracers offered no improvement with the comparison to observations. Increasing the European and/or American contributions while reducing the Asian (Siberian) contribution gave a result similar to INC_WET, and vice versa to NO_WET.  This information has been added to relevant sections of the manuscript (5.2.1, 6.2.1 and 7).*

*In the introduction we discuss the large uncertainties associated with high northern latitude wetland emissions. Given the large variability in the spatial distribution and global magnitude of emissions in both process models and inversion studies (e.g. see Table 3, Melton et al., 2013), it is hard to determine which inventory may contain the most accurate spatial distribution of emissions. Although the Fung emissions are now 25 years old, and have been proceeded by newer wetland emission estimates, it is not clear that newer estimates are necessarily better (or worse). Note that the Fung wetland emissions were used in a variety of models in the TRANSCOM studies in 2011 and 2013 (Patra et al, 2011; Saito et al., 2013).*

4. Ideally, you'd carry out a supplementary model-run in which an alternative wetland scenario is used. The Bousquet (2011) inversion inventory, for example, assimilated observations of CH4 made throughout the Arctic, and would likely, therefore, be able to capture the seasonal cycle of Arctic CH4 well. However, according to Figure 1, it does not show the same delayed seasonal cycle and large magnitude of autumnal emissions required in your FUNG_DEL cycle in order to capture the seasonal cycle of CH4 . Also, as far I can tell, it has not been compared to observations of methane isotopologues before, and doing so may back up your conclusions that significant emissions deeper into the autumn are necessary.

*Unfortunately it is not possible to perform any further comparable simulations due to changes in computer platform.*

*The Bousquet inversion inventory provides estimates of wetland emissions from 1993 to 2009, and it is the average of the years 1993 to 2004 that was shown in Figure 1 (these years were chosen originally to aid comparison to the WETCHIMP data). However, when considering the year by year data, there is a large inter-annual variation in total 50-90°N wetland spring-time emissions in the Bousquet dataset, with negative or very low total wetland emissions from latitudes >50°N occurring during May in many recent years. Years in which total May emissions >50°N are either negative or very low (similar to winter values) in the Bousquet dataset are: 2002, 2005, 2006, 2007, 2008 and 2009. Therefore emission*

*data derived from the Bousquet atmospheric methane inversions supports our result for very low high latitude wetland emissions in May (and thus a later spring/summer kick-off in wetland emissions) for the 2005-2009 period. We have now changed Figure 1 to show Bousquet average emissions from the years 2005-2009 (as these are the years that we later use for observational data).*

*As well as updating Figure 1, we have added a comment about the varying seasonality of the Bousquet dataset to Section 7.*

5. Related to this, I note that you used the GFEDv2 biomass burning inventory. Version 4 of this inventory is now available, and any changes to the impact that the heavier $\delta^{13}C$-$CH_4$ has at these locations might affect your conclusions. However, I accept that the relative contribution of biomass burning emissions compared to wetland emissions at these latitudes is probably very small and therefore unlikely to have an effect unless emissions are local to the measurements.

*We agree that ideally these simulations could be updated to use version 4 for biomass burning emissions. However, as outlined above, no further comparable model simulations are now possible. An analysis of our tagged tracers demonstrates that biomass burning emissions have a negligible impact on seasonal cycles and the latitudinal gradient at these latitudes. Therefore we do not believe updating the biomass burning emission inventory would alter our conclusions.*

6. If further simulations are not possible, I think a discussion of the effect of your choices on your results should be included in the results section.

*We assume this comment is aimed principally at our choice of wetland emission dataset (an issue also brought up by Reviewer 2). We have replied to these comments above and added further information to the manuscript about spatial/temporal emission distributions in both the Bousquet dataset (as requested by Reviewer 1) and the LPJ-Bern dataset (as requested by reviewer 2), see Section 6.2.2, Section 7.*

*In addition, in Figure 2, we have added a new panel providing more information on the latitudinal variation in emission seasonal cycles in the Fung et al. dataset. We found that varying the spatial distribution of high latitude northern wetland emissions in the Fung dataset in a simple way did not improve the comparison with atmospheric observations (please see also our reply to referee #1's comment no. 3 and referee #2's comment no. 1).*

7. Page 5, line 16: This is the first mention of the BASE scenario. You should make explicit that here, 'BASE' refers to the control experiment that uses the emissions described in the previous section, rather than some model set-up.Page 6, line 6-10: This paragraph needs a little more detail. You have not previously described the locations of those measurements made further south than Cold Bay (perhaps they could also be included in Figure 4?). You say that the gradient in δD-CH 4 is captured, and also that it is underestimated in the NH mid latitudes. Can you explain more clearly? It looks to me that perhaps the δD-CH 4 is mostly captured quite well as far to 50S, but that using the South Pole value as a baseline is shifting the model away from the observations. Perhaps it's the SH gradient that isn't captured, rather than the NH gradient?

   ***The text has been changed to read 'BASE control scenario' to make this clearer.***

   ***We agree with the referee's comments regarding the δD-CH 4 latitudinal gradient. The paragraph discussing this at the end of Section 5.1 has been expanded.***

   ***A description of the other measurement sites used in this study has been added to Section 2 and their locations plotted on the previous Figure 4 (now Figure 3).***

8. Page 8, line 12: It's a shame that there are no δD-CH 4 ratios included here for completeness, but since the changes to the wetland emissions in this section of the study don't improve simulated CH 4 or δC 13 -CH 4 concentrations, I understand the reluctance to carry out the runs.

   ***As previously mentioned, unfortunately no further comparable runs are possible at this point.***

9. Page 8, line 18: The name "WETLD_X2" is a little misleading, as emissions have been increased only by 50%. Can you change this name?

   ***We have changed the name of this scenario to INC_WETLD.***

10. Page 11, line 15: Are the model lines here full zonal means across all longitudes? If so, is there any impact on the comparisons at the sites in the Arctic if you compare only at the measurement locations? I think the plot would be too busy if you included these comparisons within it, but you could mention it in the text if there is any effect.

*Yes, these are full zonal means. We did initially include the measurement location data, but as suggested by the reviewer, the plot became too busy and so it was removed before submission. As requested, we have now added some text to Section 6.3 describing the impact of using model data from measurement station locations rather than zonal means (and to make it clear zonal means are plotted).*

11. Figure 3: I think that this plot could be a little clearer. Can you include the locations of the measurement sites here (or in Figure 4)? Is this an annual mean or is it the peak summertime emissions? Can you differentiate between regions where wetland emissions are zero and where they're just smaller than the lowest value in your colourbar?

    *We have added the location of the measurement sites to the previous Figure 4 (now Figure 3) as it enables all sites to be included. The Figure caption has been updated to clarify annual mean emissions are shown. The colourbar has been changed to differentiate regions where wetland emissions are zero.*

12. Perhaps you could include a similar second panel showing the standard deviation of the emissions, or the month during which emissions peak (or at least mention it in the text)? i.e. do emissions peak in July everywhere in the Artic, or does it vary by region?

    *We have now included a second panel in Figure 3 that shows zonally summed emissions for each of the summer emission months (seasonality does not vary greatly with longitude). For latitudes < ~70°N, emissions peak in July. For latitudes > ~70°N, emissions are fairly constant for the June-August period, and decline slightly for September.*

13. Figure 10: Can you differentiate the lines more clearly in this plot? The difference between the dash, dot, dot-dash and dot-dot-dash lines is not obvious enough in a plot of this size (especially as they only deviate in a small subsection of latitudes).

    *We have now tried using various different line types and colours for this figure. In our opinion the best improvement was obtained by swapping some of the line types in the legend and increasing the length of the y-axis.*

Technical corrections:

Page 1, line 11 and throughout: I find the use of the term 'coloured' throughout the manuscript to describe the different tracers a bit odd, although I accept that it can be

a difficult idea to describe well. I'd suggest changing to the term 'tagged' or similar for clarity.

*Text changed to read 'tagged'.*

Page 2, line 3: "to-date" -> "to date" (no hyphen)

*Changed.*

Page 12, line 29: "May-time emissions" -> "May emissions"/"emissions than predicted in May"

*Changed.*

Page 13, line 13: "currently lacking" -> "currently-lacking"

*Changed.*

*References*

*Patra, P. K., Houweling, S., Krol, M., Bousquet, P., Belikov, D., Bergmann, D., Bian, H., Cameron-Smith, P., Chipperfield, M. P., Corbin, K., Fortems-Cheiney, A., Fraser, A., Gloor, E., Hess, P., Ito, A., Kawa, S. R., Law, R. M., Loh, Z., Maksyutov, S., Meng, L., Palmer, P. I., Prinn, R. G., Rigby, M., Saito, R., and Wilson, C.: TransCom model simulations of CH4 and related species: linking transport, surface flux and chemical loss with CH4 variability in the troposphere and lower stratosphere, Atmos. Chem. Phys., 11, 12813-12837, doi:10.5194/acp-11-12813-2011, 2011.*

*Saito, R., et al., TransCom model simulations of methane: Comparison of vertical profiles with aircraft measurements, J. Geophys. Res., 118, 3891-3904, doi:10.1002/jgrd.50380, 2013.*

---

## Author Comment (AC2) · 21 Oct 2016

_**Replies to Referee #2 (bold italics).**_

General
The manuscript 'Using d13C-CH4 and dD-CH4 to constrain Arctic methane emissions' by Warwick and co-workers presents a study of atmospheric CH4 and its isotopes (d13C and dD) in the Arctic. Model simulations of all three components together with observations are used in a qualitative way to draw conclusions on the main arctic emission sources and their seasonal behavior. The analysis applies state-of-the-art modeling techniques and the methods and results are generally presented with a clear language and structure. The work clearly adds an important piece of information entangling the contributions of different methane sources and will probably help to further improve process models that are of paramount importance to understand future climate-emission feedbacks in the Arctic. I only have a few minor comments that can be addressed in the revised text but will, most likely, not require any major changes in the analysis.

***We thank the referee for their very helpful comments and suggestions which have helped improve the manuscript.***

Major comments

1.  One possibly more important remark has to do with the chosen wetland emissions in the presented simulations. Why was the rather old Feng et al. dataset chosen as a reference? As can be seen in Figure 1 the LPJ-Bern emissions already follow the suggested delay in summer-time emissions. Related to this: What is the influence of the spatial distribution of the chosen wetland emissions. Could the suggested shift in emissions also stem from an erroneous distribution in space rather than in time? How different is the spatial distribution of LPJ-Bern as compared to Feng et al?

    ***We followed the TRANSCOM model comparisons (Patra et al., 2011, Saito et al. 2013) in using the Fung et al. (1991) wetland dataset. Although there are now quite a few published wetland methane emission datasets available (see Fig. 1), uncertainties are large and it is not clear which dataset may be the most accurate or best performing.***

    ***We performed a simple analysis to investigate whether an erroneous emission distribution, rather than emission seasonality, could influence the modelled seasonality of methane mixing ratios etc. at the chosen measurement sites. Our model includes 4 wetland methane tagged tracers: north European, north American, north Asian and tropical. We tried varying the relative quantities of northern emissions (e.g. decreasing north American and/or north European while increasing Asian/Siberian emissions) and varying the relative quantities of northern vs. tropical emissions. However, the model results were very similar to either the INC_WET or NO_WET scenarios and we were unable to capture observed seasonalities in mixing ratios and/or isotopic ratios (see also our reply to referee #1's comment no. 3).***

*It is possible that inaccuracies in the emission distributions within our tagged regions could also impact modelled mixing ratio seasonalities. For example, if the model had a greater proportion of emissions >50°N located at very high latitudes, total emissions >50°N during May could be reduced (see new Fig. 2b). However, this would also have the unwanted impact of reducing emissions during October (Fig. 2b) and would alter the modelled latitudinal gradient (which is currently well captured, Fig 4). Therefore we believe it would be very difficult to correct the modelled atmospheric seasonal cycles by altering only wetland emission distributions and not seasonalities.*

*There are differences between the spatial distribution of emissions in the LPJ-Bern model and Fung et al., (1991), however the main emission hotspots are in broadly the same locations (West Siberia, Northern Europe, Hudson Bay lowlands). The differences in spatial distribution do not appear to be the cause of the different summed 50-90°N seasonalities between the 2 datasets, as the delayed seasonal cycle in the LPJ_Bern dataset relative to the Fung dataset is a consistent feature across all high latitude northern locations. In the Fung dataset, summed zonal mean September emissions are lower than corresponding emissions in the peak emission months of June, July and August, across all latitudes >50N (see Fig. 2). However, In the LPJ-Bern dataset, zonally summed June emissions are lower than corresponding emissions in the peak emission months of July, August and September, across all latitudes >50°N (not shown). We have added this information to the manuscript at the end of Section 6.2.2 (P10, l21-29).*

Minor comments

2. P1, l18/19: Clarify if by inventories you are referring to purely anthropogenic emissions here.

   *This is now clarified in the text to read 'anthropogenic or wetland emission inventories'. Whether anthropogenic or wetland emissions are implicated depends upon the seasonality and isotopic fractionation of the ESAS source.*

3. P1, l26: What is the status of the Nisbet et al. publication? If not yet published another reference is needed here.

   *The Nisbet paper has now been published in GBC. The manuscript and reference list has been updated.*

4. P2, l33: 'In this study': Does this still refer to Berchet et al. or the current study?

   *This refers to Berchet study. The text has been changed to clarify this.*

5. P3, l6/7: The given reference is rather old. Please give some newer references and a total amount of emissions here. See for example Kirschke et al. 2013 for some numbers and additional references.

   ***A newer reference (EDGAR v4.2) has been added.***

6. P3, l8: The additional stratospheric sink by Cl and O1D should be mentioned here as well, although it probably adds little to the seasonality.

   ***The stratospheric Cl and O1D sink is now listed here.***

7. P3, l22-24: You should mention the work by Rigby et al. here as well, who already ran a CH4 isotope model (both d13C and dD) to evaluate the benefits of atmospheric isotopic observations: Rigby et al. 2012, JGR, VOL. 117, D12312, doi:10.1029/2011JD017384.

   ***This study has been mentioned elsewhere in the manuscript, however we agree it would be appropriate to mention it her again and have added another reference to the Rigby study at this point.***

8. P3, l30f: In figures 5, 8 and 10 more than these 4 sites are used for comparison. Please mention which other sites are used in the figures.

   ***Further information regarding the additional measurement locations in figs 5, 8, and 10 has now been added to Section 2. Their locations are now shown in Figure 3.***

9. P4, l16-18: For which period are these values given?

   ***The period 2005 to 2009. This information has been added to the text.***

10. P5, l2: Which functional relationship was actually used to calculate the OH reaction rate coefficient? Reference or equation.

    ***Relevant references for the OH, Cl and O1D reaction rate coefficients have now been added to the manuscript in Section 4.***

11. P5, l9f: To put the importance of these other reactions into perspective, could you give an average lifetime of CH4 wrt to stratospheric and MBL reactions as well?

    ***Lifetimes for the MBL and stratospheric reactions have now been included in Section 4.***

12. P5, Table S2: Table S2 should be integrated in the main paper, since it is an essential information for the study. However, in the table it should be clarified which source is taken from which reference and how seasonality is considered.

***Table S2 has been moved into the main paper (Table 1), and each source referenced.***

13. P5, l24f: Which influence may the spin-up still have on the results. The spin-up basically leads to an initial state in which sources and sinks are in equilibrium. Is this adequate for the study period or could it be important to start from a background that is not in perfect equilibrium (due to steadily increasing emissions in reality).

*In this study we have spun up the model using anthropogenic emission data from 2005, and compared to atmospheric observations from 2005 to 2009. During the period 2000-2007, methane mixing ratios remained approximately constant in the atmosphere (excluding seasonal variations) and the global growth rate was close to zero suggesting the methane budget was approximately in equilibrium.*

*Prior to the year 2000, and post 2007, atmospheric methane levels increased, indicating a disequilibrium in the methane budget. This disequilibrium is not represented in our scenarios. Due to the long lifetime of methane, it is possible that changes is emissions pre-2000 could influence atmospheric mixing ratios post-2005, however this would be more likely to impact inter-annual methane trends than the seasonal variations considered in this paper.*

*Due to uncertainties in interannual methane emission trends, that the period considered in the paper occurs towards the end of an apparent period of equilibrium in the methane budget, and that we are considering seasonal variations rather than year to year trends, we believe that a spin up using yearly constant methane emissions is justified in this case.*

14. P6, l22: 'with the seasonal cycle'. Does this refer to the observed seasonal cycle?

*Text has been changed to read 'observed seasonal cycle'.*

15. P9, l2f: Fig. 1 should be mentioned here again, which shows the emission profiles of Fung and Fung delayed.

*A reference to figure 1 has now been included.*

16. P9, l6: 'forward': To me this is confusing. I would call it shifting the seasonal cycle backward.

*We wrote 'one month forward' as April emissions have been moved to May, May to June etc.. In order to be less confusing we have removed the phrase 'forward in the year' and instead said 'delayed by one month'.*

17. p10, l1-3: Berchet et al. 2016 clearly showed a strong seasonal cycle for emissions from ESAS, with a summertime peak in the order of what was suggested by Shakhova et al. 2012 for the whole year (see Fig. 5 in Berchet et al. 2016), i.e. 10-15 Tg/yr. In contrast, Berchet et al. suggest close to zero emissions in winter. How could this seasonality, that was not considered in the current model analysis, change the drawn conclusions concerning the impossibility to accommodate the ESAS flux as suggested by Shakhova et al.?

*We believe this point is already partially covered by summary points (a) and (b) in Section 6.3. They outline that, to accommodate a large ESAS source, our model requires a reduction in either high latitude wetland emissions or high latitude anthropogenic emissions, depending on whether ESAS emissions are considered to be seasonal or aseasonal, and the value chosen for their $\delta^{13}C$ isotopic composition.*

*A strong summertime peak for the ESAS emissions would resemble the seasonality for high latitude northern wetlands, which are also predicted to peak in the summer. Therefore, including such a seasonal cycle for ESAS emissions would make it harder to distinguish between ESAS and high latitude wetland emissions in our model simulations, particularly if ESAS were assigned a very negative $\delta^{13}C$ isotopic signature (~-70‰), similar to high latitude wetlands.*

*We have added further information to the manuscript regarding the possible impacts of a seasonal ESAS source in Section 6.3 (P12, l7-9 and P12, l30 – P13).*

18. Section 6.2: Please comment on the good agreement of the delayed temporal development used in your sensitivity run and that simulated by LPJ-Bern (as seen in Fig. 1). Does their model version already include possible CH4 emissions after freezing of the top soil?

*The WETCHIMP-WSL model intercomparison (Bohn et al. 2015) compared emissions from all the WETCHIMP models in the West Siberian region. The late peak in LPJ-Bern emissions was also identified in this study, and was found to be predominantly due to a late peak in wet mineral soil emission intensity, despite a very late peak in CH4-producing area. We have added this information to the discussion in Section 6.2.2 (final paragraph).*

19. P11, l21-23: This thought should be given some more discussion. Would it still be possible to accommodate a 50 % reduction in high-latitude anthropogenic emissions (as compared with the BASE run) within the range given by previous studies (e.g. Kirschke et al.).

*The -50% value given was an error. Anthropogenic emissions >50N total 36 Tg/yr in our scenario. If hydrate emissions are increased by 12 Tg/yr (from 5 to 17 Tg/yr), a 12Tg/yr reduction in anthropogenic emissions would be equate to a 33% reduction. This has been corrected in the text.*

*A reduction of this magnitude would remain within the range of top-down and bottom studies studies presented in Kirschke et al. review paper, although be very close to the lower estimates given for the agri-waste and fossil sources. (Biomass burning only represents a small proportion of emissions at these latitudes.) This information has been added to the manuscript in Section 6.3 (P13,l14-19).*

20. Figure 2: Should be part of the supplement. It is not essential to the study, but 'only' demonstrates that the applied model seems to perform reasonably well in terms of OH degradation.

    *Figure 2 has been moved to supplementary information.*

21. Figures 6 and 9: Why is there no observed dD for Ny-Alesund? If it was not observed there, it should be mentioned somewhere in the text.

    *dD was observed at Ny-Alesund. However after quality control, there was not sufficient data to be able to plot a seasonal cycle. This has now been explained in the caption for Fig. 6.*

22. Adding a table summarizing all model runs (BASE and sensitivities) and their specific settings would be nice. All the sensitivities should be given an abbreviation/name (not done in all cases) in this table and in the text so that it is easier to quickly identify what the specifics of a certain run

    *A Table summarising all model runs has now been added to the manuscript (Table 2).*

**References**

*Patra, P. K., Houweling, S., Krol, M., Bousquet, P., Belikov, D., Bergmann, D., Bian, H., Cameron-Smith, P., Chipperfield, M. P., Corbin, K., Fortems-Cheiney, A., Fraser, A., Gloor, E., Hess, P., Ito, A., Kawa, S. R., Law, R. M., Loh, Z., Maksyutov, S., Meng, L., Palmer, P. I., Prinn, R. G., Rigby, M., Saito, R., and Wilson, C.: TransCom model simulations of CH4 and related species: linking transport, surface flux and chemical loss with CH4 variability in the troposphere and lower stratosphere, Atmos. Chem. Phys., 11, 12813-12837, doi:10.5194/acp-11-12813-2011, 2011.*

*Saito, R., et al., TransCom model simulations of methane: Comparison of vertical profiles with aircraft measurements, J. Geophys. Res., 118, 3891-3904, doi:10.1002/jgrd.50380, 2013.*

---

## Author Response (AR1)

**Using $\delta^{13}$ C-CH4 and $\delta$ D-CH4 to constrain Arctic methane emissions**

Nicola J. Warwick1,2, Michelle L. Cain1, Rebecca Fisher3, James L. France4, David Lowry3, Sylvia E. Michel5, Euan G. Nisbet3, Bruce H. Vaughn5, James W. C. White5 and John A. Pyle1,2

1National Centre for Atmospheric Science, NCAS, UK.

[revised manuscript text omitted]
 CH4 + OH reaction would result in  $\delta^{13}$ C-CH4 and  $\delta$ D-CH4 seasonal cycles 180° out of phase with the CH4 seasonal cycle: the minimum in CH4 mixing ratio corresponding to maxima in  $\delta^{13}$ C-CH4 and  $\delta$ D-CH4. However, phase relationships between observed seasonal cycles in CH4,

15  $\delta^{13}$ C-CH4 and  $\delta$ D-CH4 are also influenced by seasonal variations in surface sources and lesser, alternate sinks leading to more complicated phase relationships.

The reaction of CH3D with OH has a larger KIE than the reaction of 13CH4 with OH (see Table S1). Therefore seasonal variations in atmospheric  $\delta$ D-CH4 will tend to be more dominated by seasonal changes in the OH sink than  $\delta$ 13C-CH4, with atmospheric  $\delta$ 13C-CH4 being relatively more influenced by sources. Figure 5 shows that the observed seasonal cycle of  $\delta$ D-

[revised manuscript text omitted]
|------------------------------------------------------------------------------------------------------------------------------|-------------------------|-----------------------|--------------------------------------------|------------------------------|--|
| Surface Source/Sink                                                                                                          | Global flux      | High latitude (>50°N) | $\underline{\delta^{13}\text{C-CH}_4}(\%)$ | δD-CH4 (‰) |  |
|                                                                                                                              | (Tg/yr)          | flux (Tg/yr)   |                                            |                              |  |
| Northern Wetlands                                                                                                            | $30^{1}$                | 30.0           | -70 d,h,1,n*                    | -360f,n*   |  |
| Tropical Wetlands                                                                                                            | $200^{1}$               | 0.0            | -55b,m*                  | -320g,o*   |  |
| Hydrates                                                                                                                     | $\underline{5^1}$       | 5.0            | -55d*                    | -190p      |  |
| Coal                                                                                                                  | $40^{2,3}$              | 3.2            | -50i,q*                  | -140p      |  |
| Gas                                                                                                                          | 632,3 | 15.3           | $-40^{b,n^*}$                              | -185i,j,n* |  |
| Biomass burning                                                                                                              | 314   | 3.1            | -26b                     | -210k      |  |
| Ruminants                                                                                                             | $110^{2}$               | 8.0            | -63a,c*                  | -360a*     |  |
| Landfills                                                                                                                    | $27^{2}$                | 4.6            | -53b                     | -310i,p    |  |
| Sewage                                                                                                                | $29^{2}$                | 1.8            | -57b                     | -310r      |  |
| Rice                                                                                                                  | $33^{2}$                | 0.0            | -62b,g,m*                | -330p*     |  |
| Termites                                                                                                              | $20^{1}$                | 1.1            | -57e,m*                  | -390p      |  |
| Total                                                                                                                        | 588              | 72.1           |                                            |                              |  |
| The geographical and seasonal distribution of methane flux data is based on 1 Fung et al., 1991, 2 EDG |                         |                       |                                            |                              |  |

disotonio signati 1.1 TOMONT

AR v4.1 (http://edgar.jrc.ec.europa.eu/overview.php?v=41) for 2005, 3Gurney et al., 2005, and 4Van der Werf et al., 2006. Source isotopic signature data are based on reported values from: aBilek et al., 2001, bDlugokencky et al., 2011, cLevin et al., 1993, dFisher et al., 2011, eGupta et al., 1996, fNakagawa et al., 2002a, gNakagawa et al., 2002b, hO'Shea et al., 2014, iOuay et al., 1999, jSchoell, 1980, kSnover et al., 2000, lSriskantharajah et al., 2012, mTyler et al., 1988, nUmezawa et al., 2012, oWaldron et al., 1999, pWhiticar and Schaefer, 2007, qZazzeri et al., 2015, rvalue used taken from landfill data, \*value is within a range of quoted literature estimates.

15

| Table 2.            |                                                                                                                                  |
|----------------------------|----------------------------------------------------------------------------------------------------------------------------------|
| Scenario            | Difference from BASE Scenario                                                                                                    |
|                            |                                                                                                                                  |
| BASE                       | ±                                                                                                                                |
| DEC_KIE                    | $\underline{\text{KIE}}^{(\text{CH4+OH})}/\underline{\text{KIE}}^{(\text{CH3D+OH})}$ is decreased from 1.29 to 1.16 a |
| WETLD_δD            | $\delta D$ signature for wetland emissions >50°N changed to -500‰                                                                |
| NO_WETLD                   | Wetland emissions >50°N removed                                                                                                  |
| INC_WETLD                  | Wetland emissions >50°N increased by 50% to 45 Tg/yr                                                                             |
| DEL WET                    | Seasonal cycle of wetland emissions >50°N delayed by one month throughout the year                                               |
| NO HYD              | Hydrate emissions removed                                                                                                        |
| INC_HYD                    | Hydrate emissions increased to 17 Tg/yr                                                                                          |
| WET_HYD                    | Hydrate emissions increased to 17 Tg/yr and wetland emissions decreased to 18 Tg/yr                                              |
| WET HYD δ13C        | As WET HYD, except isotopic signature for ESAS emissions is changed to -70 ‰                                                     |
| a See Table S1. |                                                                                                                                  |